# Certified Robustness on Visual Graph Matching via Searching Optimal Smoothing Range

## Abstract

Deep visual graph matching (GM) involves finding a permutation matrix that indicates the correspondence between keypoints from a pair of images and their associated keypoint positions. Recent empirical studies have shown that visual GM is susceptible to adversarial attacks, which can severely impair the matching quality and jeopardize the reliability of downstream applications. To our best knowledge, certifying robustness for deep visual GM remains an open challenge, which entails addressing two main difficulties: how to handle the paired inputs and the large permutation output space, and how to balance the trade-off between certified robustness and matching performance. In this paper, we propose a method, Certified Robustness based on Optimal Smoothing Range Search (CR-OSRS), which provides a robustness guarantee for deep visual GM, inspired by the random smoothing technique. Unlike the conventional random smoothing methods that use isotropic Gaussian distributions, we build the smoothed model with a joint Gaussian distribution, which can capture the structural information between keypoints and mitigate the performance degradation caused by smoothing. We design a global optimization algorithm to search the optimal joint Gaussian distribution that helps achieve a larger certified space and higher matching performance. Considering the large permutation output space, we partition the output space based on similarity, which can reduce the computational complexity and certification difficulty arising from the diversity of the output matrix. Furthermore, we apply data augmentation and a similarity-based regularization term to enhance the smoothed model performance during the training phase. Since the certified space we obtain is high-dimensional and multivariable, it is challenging to evaluate directly and quantitatively, so we propose two methods (sampling and marginal radii) to measure it. Experimental results on GM datasets show that our approach achieves state-of-the-art $\ell_2$ certified robustness. The source codes will be made publicly available.

## 1 Introduction

As an essential and popular combinatorial optimization problem, graph matching (GM) has attracted lasting and wide attention over the decades with also wide applications in vision (Wang et al., 2019), text (Xiong et al., 2019), graphics (Vladimir et al., 2012), pattern recognition (Vento, 2015), and machine learning (Zhang et al., 2022) etc. Meanwhile, studies on the robustness of machine learning models have attracted intensive attention, while the robustness of combinatorial solvers remains a crucial, yet largely unexplored area (Geisler et al., 2021; Lu et al., 2021). Under the deep visual GM paradigm, Ren et al. (2022) show that visual GM algorithms are vulnerable to perturbations added to keypoints and image pixels, and propose an empirical defense algorithm based on an appearance-aware regularizer. However, there is still a lack of a principled certified defense to provide theoretical robustness guarantees for GM (let alone other combinatorial problems). Certified robustness and empirical robustness are two distinct concepts in the context of adversarial machine learning. Certified robustness provides a rigorous verification of the model's output invariance under a bounded perturbation set, regardless of the attacks employed. Empirical robustness, however, lacks such a theoretical guarantee and only evaluates the model's defense capabilities against existing attack methods, which may not generalize to future unseen attacks. In fact, existing certified robustness mechanisms (including randomized smoothing, which we focus on in this study) in the graph domain (Bojchevski et al., 2020; Jia et al., 2020; Rong et al., 2019; Zügner & Günnemann, 2020)

are limited to the unconstrained node-level or graph-level classification or prediction task within a single graph, which cannot be easily adopted to solve cross-graph and combinatorial problems with structured output, such as the permutation matrix in GM.

In general, certified robustness aims to design solvers whose prediction for any input $x$ is verifiable invariant within some set around the input (Wong & Kolter, 2018). Randomized smoothing (RS) (Lecuyer et al., 2019; Cohen et al., 2019) is a promising approach to achieve certified defense of large-scale neural networks against arbitrary attacks. Given an input $x$ and a base function, RS constructs a smoothed function that is certifiably robust within the region defined by $x$ and the smoothing distribution $\mathcal{D}$ (usually an isotropic Gaussian distribution). RS has been widely applied to certify various models, e.g., image classification (Yang et al., 2020) and object detection in vision (Chiang et al., 2020), which motivates us to develop RS-based certified robustness for visual GM.

Applying RS to visual GM poses several challenges. **C1: paired inputs.** The input of visual GM consists of paired images and keypoint position matrices, which means that perturbations are also in pairs and mutually constrained in the certified space. This differs from the single input setting of previous certification problems. **C2: dependency of keypoints.** The graph structure derived from Delaunay triangulation of keypoint positions as a whole conveys important structural information and is an essential intermediate result for the visual GM model, which motivates us to preserve the original graph structure during the smoothing process to maintain the matching performance. **C3: large permutation output space.** The output of visual GM is a 0-1 permutation matrix, which has an exponential number of theoretical possibilities. For a matching task with $n$ keypoints, the output is an $n \times n$ matrix, and there are $n!$ theoretically possible outputs. This means that we cannot directly apply the existing RS definition, which assumes that a visual GM task is a classification problem, and estimate the occurrence probability for each possible output. This would cause a computational explosion. **C4: performance degradation caused by smoothing.** Smoothing can affect model performance, as evidenced by previous studies. Although data augmentation is a conventional method to improve performance, it is not designed for visual GM and its effect is unsatisfactory.

To address these challenges, we propose **C**ertified **R**obustness based on **O**ptimal **S**moothing **R**ange **S**earch (**CR-OSRS**), a novel robustness certification method for visual GM. Specifically, CR-OSRS assumes that the two perturbations within the pair belong to the joint input space and derives a certification result that respects the inter-pair constraints (**C1**). CR-OSRS also designs a smoothed model with a joint Gaussian distribution that captures the correlation of keypoints and uses global optimization to determine the optimal correlation parameters that enhance certified robustness. The rationale of this design is to preserve the difference and avoid the confusion of keypoints under perturbations as much as possible (**C2**). Furthermore, CR-OSRS operates on a subspace in the output space determined by a similarity threshold and defines the certified robustness as the output always within the subspace under perturbations. This eliminates the need to count the probability of each possible output and only requires calculating the probability that the output falls into the subspace (**C3**). Additionally, CR-OSRS proposes a data augmentation method for GM using joint Gaussian noise and an output similarity-based regularization term, which improves both the matching accuracy and certified robustness (**C4**).

**The contributions of this paper are as follows: (1)** We propose a novel certification method for visual GM, CR-OSRS, that provides the rigorous robustness guarantee by characterizing a certified $\ell_2$ norm space (see Theorem 4.1). This robustness means that when the perturbation is within the certified input space, the smoothed model always predicts the output within the output subspace. **(2)** Specifically, we propose to use the joint Gaussian distribution to build a smoothed model and globally optimize the correlation parameters in the distribution. This method can capture the connection of keypoints to enhance the anti-disturbance ability of the model (see Sec. 4.2). We also apply data augmentation with joint Gaussian noise and the output similarity-based regularization term during the training phase to further improve the model performance (see Sec. 4.3). **(3)** We propose two methods, sampling and marginal radii respectively, to measure the certified space for quantitative analysis (see Sec. 4.4). We evaluate our approach on the Pascal VOC dataset (Everingham et al., 2010) with Berkeley annotations (Bourdev & Malik, 2009), the Willow ObjectClass dataset (Cho et al., 2013) and SPair-71k dataset (Min et al., 2019) for six representative GM solvers. The results show that CR-OSRS can provide robustness guarantees for visual GM, and the CR-OSRS mechanism performs better than directly applying RS (Cohen et al., 2019) to visual GM, which we refer to as RS-GM. Moreover, the training methods we designed are also effective (see Sec. 5).

## 2 RELATED WORKS

We review studies on certified robustness through RS and discuss the robustness of GM. To our knowledge, this is the first study to combine the RS and GM communities.

**Randomized Smoothing based Certified Robustness.** RS is proposed in Lecuyer et al. (2019) as a certified adversarial defense and used to train the (first) certifiably robust classifier on ImageNet. However, it provides loose guarantees. Cohen et al. (2019) show that Gaussian noise addition provides a tight $\ell_2$ certification radius, with subsequent works on new RS-type techniques, e.g. techniques using smoothing distributions at different norms (Levine & Feizi, 2021; Lee et al., 2019; Yang et al., 2020), and techniques for different tasks (Chiang et al., 2020; Jia et al., 2020; Kumar & Goldstein, 2021). However, all previous smoothing distributions $\mathcal{D}$ deprive of favorable prior knowledge, which mainly refers to the keypoint positions and graph structure in visual GM. Moreover, all of them only certify a single image or graph but do not consider the combinatorial nature as in visual GM.

**Graph Matching and its Robustness.** Approximate GM solvers have evolved from traditional methods without learning (Emmert-Streib et al., 2016) to learning-based ones (Yan et al., 2020). Seminal work (Zanfir & Sminchisescu, 2018) proposes a deep neural network pipeline for visual GM, in which image features are learned through CNN, with subsequent variants (Rolínek et al., 2020; Wang et al., 2019), among which a major improvement is to exploit structural information using different techniques, for example GNN, rather than only using appearance for node/edge attributes as done in Zanfir & Sminchisescu (2018). Our work, which uses the RS-type technique, treats the GM solver as a black box irrespective of whether it is learning-based or not.

There are also works on adversarial attacks and defense on (deep) GM. Previous work (Yu et al., 2019b) proposes a robust graph matching (RGM) model against perturbations, e.g., distortion, rotation, outliers, and noise. Zhang et al. (2020) devise an adversarial attack model for deep GM networks, which uses kernel density estimation to construct dense regions such that neighboring nodes are indistinguishable. Ren et al. (2021) devise two specific topology attacks in GM: inter-graph dispersion and intra-graph combination attacks, and propose a resilient defense model. Lin et al. (2023) integrate the momentum distillation strategy to balance the quadratic contrastive loss and reduce the impact of bi-level noisy correspondence. However, these defense methods are all heuristic and lack robustness certification against unseen attacks.

## 3 PRELIMINARIES

**Randomized Smoothing.** The original RS (Cohen et al., 2019) can transform an arbitrary base classifier $f$ into a smoothed classifier $g$ that is certifiably robust under $\ell_2$ norm. For any single input $x$, the smoothed classifier $g$ returns the most probable prediction of $f$ for the random variable $\mathcal{N}(x; \sigma^2 I)$, which is defined as:

$$g(x) = \arg\max_{c \in \mathcal{Y}} P(f(x + \varepsilon) = c), \qquad (1)$$

where $\varepsilon \sim \mathcal{N}\left(0, \sigma^2 I\right)$ is an isotropic Gaussian noise. Then the certified radius within which the output is unchanged for $g(x + \delta) = c_A$ that measures the certified robustness is:

$$\|\delta\|_2 < \frac{\sigma}{2}\left(\Phi^{-1}\left(\underline{p_A}\right) - \Phi^{-1}\left(\overline{p_B}\right)\right), \qquad (2)$$

where the most probable class $c_A$ is returned with probability $p_A$ and the "runner-up" class is returned with probability $p_B$. $\underline{p_A}$ and $\overline{p_B}$ are the lower bound and upper bound of $p_A$ and $p_B$, respectively, and $\Phi^{-1}$ is the inverse of the standard Gaussian cumulative distribution function.

**Visual Graph Matching.** We consider the visual GM task $f$ which is a comprehensive setting allowing for both visual appearance and structural perturbation: $(\mathbf{c}^1, \mathbf{c}^2, \mathbf{z}^1, \mathbf{z}^2) \to \mathbf{X}$, where $(\mathbf{c}^1, \mathbf{c}^2)$ is the image pair with keypoint position pair $(\mathbf{z}^1 \in \mathbb{R}^{n_1 \times 2}, \mathbf{z}^2 \in \mathbb{R}^{n_2 \times 2})$, $\mathbf{X} \in \{0, 1\}^{n_1 \times n_2}$ represents a 0-1 permutation matrix, $n_1$ and $n_2$ are the numbers of keypoints. Recent deep GM methods tackle images with keypoints as inputs in an end-to-end manner (Rolínek et al., 2020; Wang et al., 2019; 2021; Zanfir & Sminchisescu, 2018) and typically comprise three components: keypoint feature extractor, affinity learning, and final correspondence solver. First, two graphs $\mathcal{G}_1$ and $\mathcal{G}_2$ are constructed by Delaunay triangulation (Lee & Schachter, 1980). Then node features are obtained via a feature extractor based on the keypoint positions. Afterward, edge features are constituted based on node

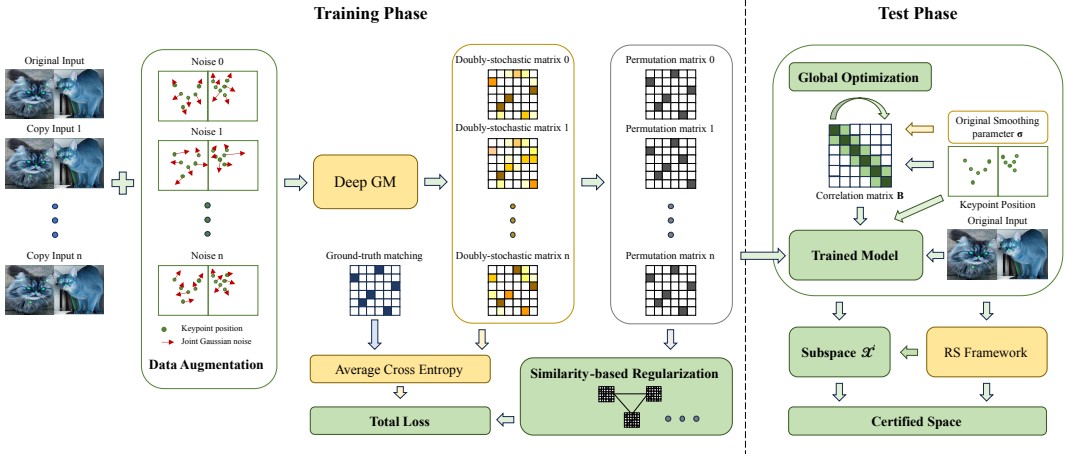

Figure 1: The pipeline of this work consists of two phases: training and testing. In the training phase, we enhance the provable robustness and matching accuracy of the model simultaneously by applying data augmentation and a regularization term as defined in Eq. 11. In the testing stage, we first construct a joint Gaussian distribution and use the global optimization in Eq. 10 to search for the optimal smoothing range. Moreover, we use the optimization results and the trained model to construct a smooth model, and then compute the output subspace and input certified space as described in Sec. 4.1. Fig. 1 illustrates an example of using the NGMv2 (Wang et al., 2021) solver and finding a robustness guarantee under keypoint position perturbations.

features and topology information of $\mathcal{G}_1$ and $\mathcal{G}_2$. Based on these node and edge features, the affinity matrix $\mathbf{K} \in \mathbb{R}^{n_1 n_2 \times n_1 n_2}$ is initialized which is then fed to the affinity learning layer to learn the node-to-node and edge-to-edge correspondence similarity. Finally, the correspondence solver outputs the predicted permutation matrix $\mathbf{X}$ by solving quadratic assignment problem (QAP) (Loiola et al., 2007) which aims to maximize the overall affinity score $J$:

$$\max_{\mathbf{X}} J(\mathbf{X}) = \mathrm{vec}(\mathbf{X})^\top \mathbf{K}\, \mathrm{vec}(\mathbf{X}),$$
$$\text{s.t. } \mathbf{X} \in \{0,1\}^{n_1 \times n_2}, \mathbf{X}\mathbf{1}_{n_1} = \mathbf{1}_{n_1}, \mathbf{X}^\top \mathbf{1}_{n_2} \leq \mathbf{1}_{n_2}, \tag{3}$$

where $\mathrm{vec}(\mathbf{X})$ denotes the column-wise matrix of $\mathbf{X}$ which is a partial permutation matrix if $n_1 < n_2$.

As discussed above, image pixels affect the extracted node and edge features, while keypoint positions affect the extracted node features by influencing the bilinear interpolation and the graph structure extracted by Delaunay triangulation. However, the keypoint positions are inherently vulnerable due to the randomness of human labeling or keypoint detectors (which are used in the pre-processing step to locate key objects in an image (Bourdev & Malik, 2009)), and the image pixels are extremely sensitive to various noises as in other image tasks. Therefore, in this study, we consider the robustness of visual GM under two types of perturbations: image pixels and keypoint positions as in Ren et al. (2022). As these two perturbations belong to different spaces and exhibit different effects on GM models, we devise different certification schemes for them. We investigate the certified robustness against attacks on image pixels while holding the keypoint positions constant, and attacks on keypoint positions while holding the image pixels constant.

## 4 METHODOLOGY

This section introduces the methodology of this work that comprises four parts: (1) the definition of a smoothed model and the theoretical framework developed for certified robustness analysis (Sec. 4.1); (2) the construction of the joint Gaussian distribution and an optimization method that helps to find the optimal correlation parameter to balance the trade-off between certification validity and model performance (Sec. 4.2); (3) a training method that incorporates data augmentation with joint Gaussian noise and an output similarity-based regularization term that constrains the smoothed output gaps (Sec. 4.3); (4) two methods for quantifying the certified space, sampling and marginal radii, respectively (Sec. 4.4). The pipeline is shown in Fig. 1 with the process detailed in Alg. 1.

### 4.1 ROBUSTNESS GUARANTEE FOR VISUAL GRAPH MATCHING

As discussed in Sec. 3, we certify the robustness under two types of perturbations: keypoint position perturbations and image pixel perturbations respectively. In this subsection, we focus on the

certified robustness under keypoint position perturbations, and the certified robustness under image perturbations can be derived in a similar way.

As stated in Sec. 1, visual GM poses a challenge for certified robustness due to its large permutation output space. Previous line of research e.g. Cohen et al. (2019) aim to certify that the output remains unchanged under perturbations, but this may result in a failed certification or a small certification range for visual GM due to the lack of a dominant matrix, that is, the probability difference between the most probable matrix and the "runner-up" matrix is small. Furthermore, it is computationally intractable to enumerate the probabilities of all possible matrices. Therefore, we propose a novel certified robustness definition that guarantees the output always belongs to an output subspace centered on the core output.

We first define a core output $\mathbf{X}_c$. When queried at $(\mathbf{c}^1, \mathbf{c}^2, \mathbf{z}^1, \mathbf{z}^2)$, $\mathbf{X}_c$ is a more likely output of base GM function $f$ when $(\mathbf{z}^1, \mathbf{z}^2)$ is perturbed by joint Gaussian noise:

$$\mathbf{X}_c = H(S(\sum f\left(\mathbf{c}^1, \mathbf{c}^2, \mathbf{z}^1 + \varepsilon_1, \mathbf{z}^2 + \varepsilon_2\right))),$$
$$\text{where } \varepsilon_1 \sim \mathcal{N}\left(0, \mathbf{\Sigma_1}\right), \varepsilon_2 \sim \mathcal{N}\left(0, \mathbf{\Sigma_2}\right), \tag{4}$$

where the smoothing noise $\varepsilon_1$ and $\varepsilon_2$ follow joint Gaussian distributions with covariance matrices $\mathbf{\Sigma_1}$ and $\mathbf{\Sigma_2}$, which represent constraints between keypoints $\mathbf{z}^1$ and $\mathbf{z}^2$ respectively (for solving **C1**). $S$ is the Sinkhorn operator that converts the vertex score matrix into a doubly-stochastic matrix and $H$ is the Hungarian operator that transforms a doubly-stochastic matrix into a permutation matrix. The computation of Eq. 4 takes into account the "majority decision" of the smoothed model while only needing to save the sum of matching matrices rather than the statistics of each possible matrix. Note that $\mathbf{X}_c$ is not the output we want to certify; it is just the center point of the subspace to be constructed, and so there is no need to consider whether this computation process is provably robust.

Next, we define a subspace $\mathcal{X}'$ of the entire output space $\mathcal{X}$ by a similarity threshold $s \in [0, 1]$, and the similarity between the elements in $\mathcal{X}'$ and the core output $\mathbf{X}_c$ is no less than $s$ (for solving **C3**).

$$\mathcal{X}' = \left\{ \mathbf{X}_i \left| \frac{\mathbf{X}_i \cdot \mathbf{X}_c}{\mathbf{X}_c \cdot \mathbf{X}_c} \geq s, \mathbf{X}_i \in \mathcal{X} \right. \right\}, \tag{5}$$

where we employ a simple dot product $\mathbf{X}_i \cdot \mathbf{X}_c$ to measure the number of identical matching keypoints in these two output matrices, because the output matrices are 0-1 permutation matrices. Similarly, $\mathbf{X}_c \cdot \mathbf{X}_c$ calculates the total number of keypoints to be matched.

According to the above definition, we construct a new base function $f_0$ based on $f$. Specifically, we partition the entire output space into two parts according to Eq. 5, then assign all points inside $\mathcal{X}'$ with 1 and the rests with 0, and finally convert $f$ to a binary function $f_0$:

$$f_0\left(\mathbf{c}^1, \mathbf{c}^2, \mathbf{z}^1, \mathbf{z}^2\right) = \left\{ \begin{array}{l} 1, \text{ if } f(\mathbf{c}^1, \mathbf{c}^2, \mathbf{z}^1, \mathbf{z}^2) \in \mathcal{X}' \\ 0, \text{ otherwise} \end{array} \right. . \tag{6}$$

Then we build a smoothed function $g_0$ from $f_0$. When queried at the input $\left(\mathbf{c}^1, \mathbf{c}^2, \mathbf{z}^1, \mathbf{z}^2\right)$ with fixed $(\mathbf{c}^1, \mathbf{c}^2)$, $g_0$ outputs the binary labels when $(\mathbf{z}^1, \mathbf{z}^2)$ is perturbed by joint Gaussian noise:

$$g_0\left(\mathbf{c}^1, \mathbf{c}^2, \mathbf{z}^1, \mathbf{z}^2\right) = \left\{ \begin{array}{l} 1, \text{ if } P(f(\mathbf{c}^1, \mathbf{c}^2, \mathbf{z}^1 + \varepsilon_1, \mathbf{z}^2 + \varepsilon_2) \in \mathcal{X}') \geq 1/2 \\ 0, \text{ otherwise} \end{array} \right. , $$
$$\text{where } \varepsilon_1 \sim \mathcal{N}\left(0, \mathbf{\Sigma_1}\right), \varepsilon_2 \sim \mathcal{N}\left(0, \mathbf{\Sigma_2}\right). \tag{7}$$

**Theorem 4.1** ($\ell_2$ **norm certified space for visual GM**). *Let $f$ be a matching function, $f_0$ and $g_0$ be defined as in Eq. 6 and Eq. 7, $\varepsilon_1 \sim \mathcal{N}\left(0, \mathbf{\Sigma_1}\right), \varepsilon_2 \sim \mathcal{N}\left(0, \mathbf{\Sigma_2}\right)$. Suppose $\underline{p} \in (\frac{1}{2}, 1]$ satisfy:*

$$P(f_0\left(\mathbf{c}^1, \mathbf{c}^2, \mathbf{z}^1 + \varepsilon_1, \mathbf{z}^2 + \varepsilon_2\right) = 1) = P(f(\mathbf{c}^1, \mathbf{c}^2, \mathbf{z}^1 + \varepsilon_1, \mathbf{z}^2 + \varepsilon_2) \in \mathcal{X}') = p \geq \underline{p}. \tag{8}$$

*Then we obtain the $\ell_2$ norm certified space for the noise pair $(\delta_1, \delta_2)$:*

$$\frac{\delta_1^T \mathbf{\Sigma_1}^{-1} \delta_1 + \delta_2^T \mathbf{\Sigma_2}^{-1} \delta_2}{\|\delta_1^T \mathbf{\Sigma_1}^{-1} \mathbf{B_1} + \delta_2^T \mathbf{\Sigma_2}^{-1} \mathbf{B_2}\|} < \Phi^{-1}\left(\underline{p}\right), \tag{9}$$

*which guarantees $g_0\left(\mathbf{c}^1, \mathbf{c}^2, \mathbf{z}^1 + \delta_1, \mathbf{z}^2 + \delta_2\right) = 1$. $\mathbf{B}_1 \in \mathbb{R}^{n_1 \times n_1}$ and $\mathbf{B}_2 \in \mathbb{R}^{n_2 \times n_2}$ are full rank and real symmetric matrices based on the keypoint correlation in keypoint position matrices $\mathbf{z}^1$ and $\mathbf{z}^2$, satisfying $\mathbf{B}_1^\top \mathbf{B}_1 = \mathbf{\Sigma}_1$ and $\mathbf{B}_2^\top \mathbf{B}_2 = \mathbf{\Sigma}_2$.*

Finally, we formulate a robustness guarantee of $g_0$ that ensures the similarity between the matching matrix and $\mathbf{X}_c$ being no less than $s$, that is, the matching matrix always belongs to the subspace $\mathcal{X}'$. We present and illustrate the detailed settings as well as the properties of $\mathbf{B}_1$ and $\mathbf{B}_2$ in Sec. 4.2. The complete proof of Theorem 4.1 is provided in Appendix A.

## 4.2 Joint Smoothing Distribution

This subsection presents the detailed settings and properties of $\mathbf{B}_1$ and $\mathbf{B}_2$ under keypoint position perturbations. Additionally, we introduce an optimization method to search the optimal smoothing range for enhanced robustness. Besides, refer to Appendix C.2 for the case under pixel perturbations.

As stated in Sec. 3, keypoint positions influence the extracted features through bilinear interpolation and directly determine the graph structure derived by Delaunay triangulation. If the smoothing noise for each keypoint position is completely independent, then the perturbed keypoint set may exhibit partial overlaps or high similarities. This may cause the extracted features to overlap and thus degrade the matching performance. Therefore, our objective is to design a smoothing distribution that can preserve the diversity of keypoints (for solving **C2**).

To construct the correlation matrices $\mathbf{B}_1$ and $\mathbf{B}_2$, we use a correlation parameter $b$. The diagonals of $\mathbf{B}_1$ and $\mathbf{B}_2$ are original $\sigma$ as in RS (Cohen et al., 2019), the off-diagonal elements adjacent to the main diagonal are $\sigma \times b$, and the remaining elements are 0. This setting not only maintains the correlation between keypoints but also allows $b$ and $\sigma$ to be global parameters that can be optimized. We devise an optimization algorithm that aims to maximize the volume of the certified space through the proxy radius, which will be defined in Sec. 4.4. We impose a constraint on $b$ in the optimization algorithm to keep it within a reasonable range, as a large $b$ may enhance the matching performance but diminish the certified space. The optimization problem can be written as:

$$\arg\max_{\sigma,b} \Phi^{-1}\left(\underline{p}\right) \sum_{i=1,2}\left(\sqrt[2m_i]{\prod_{j}^{m_i}\lambda_{ij}}\right) + \kappa b, \tag{10}$$

where $\kappa \in \mathbb{R}^+$ is a hyperparameter, $\lambda_{ij}$ is the $j$-th eigenvalue of $\mathbf{\Sigma}_i$, and $m_i$ is the eigenvalue number of $\mathbf{\Sigma}_i$. This optimization idea is inspired by the framework in Alfarra et al. (2022); Eiras et al. (2021), but the main difference is that their optimization is for individual input test points, while our optimization method is a global optimization for the whole data set. Therefore, our method does not suffer from the data independence problem in Alfarra et al. (2022); Eiras et al. (2021).

## 4.3 Training Phase with Data Augmentation and an Output Similarity-based Regularization Term

As noted in the previous RS method (Lecuyer et al., 2019; Cohen et al., 2019), the smoothing noise influences the matching performance. To improve both the matching performance and the provable robustness, we adopt two strategies (for solving **C4**). The first one is data augmentation, which is motivated by Cohen et al. (2019). The difference is that we use a joint Gaussian distribution for data augmentation, which is consistent with the type of distribution we used to construct the smoothed model. The second one is a regularization term based on the similarity between smoothed outputs. Since RS has the property of "majority decision", minimizing the loss between each output and the true matching result is not adequate. We also need to ensure that outputs under multiple perturbations are as consistent as possible for a fixed datum. Therefore, we copy the same datum $n$ times, perform data augmentation on the $n$ data, and compute their corresponding outputs, then add a regularization term to penalize the divergence between $n$ outputs. The total loss function can be written as follows:

$$\mathcal{L} = \frac{1}{n}\sum_{i}^{n}\mathcal{L}_{GM}\left(\mathbf{X}_i, \mathbf{X}_{gt}\right) + \beta\sum_{i,j}^{n}(1 - \frac{\mathbf{X}_i \cdot \mathbf{X}_j}{\mathbf{X}_{gt} \cdot \mathbf{X}_{gt}}), \tag{11}$$

where $\beta \in \mathbb{R}^+$ is a hyperparameter, $\mathbf{X}_{gt}$ is the true matching for input $(\mathbf{c}^1, \mathbf{c}^2, \mathbf{z}^1, \mathbf{z}^2)$, $\mathbf{X}_i$ and $\mathbf{X}_j$ are the outputs for $f(\mathbf{c}^1, \mathbf{c}^2, \mathbf{z}^1 + \varepsilon_1, \mathbf{z}^2 + \varepsilon_2)$ when $(\varepsilon_1, \varepsilon_2)$ are sampled by the $i$-th and $j$-th times, respectively. $\mathcal{L}_{GM}$ is the original loss function in GM methods, such as binary cross-entropy (Wang et al., 2021) and pixel offset regression (Zanfir & Sminchisescu, 2018). In Eq. 11, the first term represents the average matching loss, which uses data augmentation based on the joint Gaussian distribution to improve the matching accuracy under perturbations. The second regularization term

imposes a similarity constraint between the outputs, which will help increase $p$ in Eq. 8 and enhance the provable robustness.

## 4.4 QUANTIFY CERTIFICATION

In Sec. 4.1, we derive Eq. 9 to characterize the certified space with multiple perturbations, which is, however, challenging to quantify and compare. Moreover, previous studies have not tackled the problem of certification with multiple perturbations. To address this issue, we propose two quantity metrics to measure certified robustness: sampling and marginal radii.

**Sampling.** According to Eq. 9, the certified robustness of $g_0$ increases when the certified space becomes larger, which means that more pairs of $(\delta_1, \delta_2)$ satisfy the certified space. However, it is impractical to determine how many and which pairs of $(\delta_1, \delta_2)$ satisfy Eq. 9, so we propose using a sampling approach to approximate certified robustness. Specifically, we sample the noise pairs $(\delta_1, \delta_2)$ from the distributions and verify if they satisfy Eq. 9. The approximate certified robustness of $g_0$ is given by the probability of sampled noises that satisfy Eq. 9.

**Marginal Radii.** Moreover, by fixing one of $\delta_1$ and $\delta_2$, we can simplify the joint space of Eq. 9 to a marginal space, which facilitates robustness evaluation. Specifically, we set one of $\delta_1$ and $\delta_2$ to be a zero matrix and derive a simpler expression for Eq. 9 as follows. As an example, we consider the case of setting $\delta_2$ to a zero matrix:

$$\|\delta_1^\top \mathbf{B}^{-1}\| < \left(\Phi^{-1}\left(\underline{p}\right)\right). \tag{12}$$

**Lemma 4.2** (Eigenvalue Comparison). *For a real symmetric matrix $\mathbf{A} \in \mathbb{R}^{n \times n}$, with $\lambda_{max}$ and $\lambda_{min}$ as its maximum and minimum of eigenvalues, then $\forall \mathbf{X} \in \mathbb{R}^n, \lambda_{min}\mathbf{X}^\top\mathbf{X} \leq \mathbf{X}^\top\mathbf{A}\mathbf{X} \leq \lambda_{max}\mathbf{X}^\top\mathbf{X}$.*

Using Lemma 4.2, we know that $\frac{1}{\lambda_{1\max}}\delta_1^\top\delta_1 \leq \delta_1^\top\mathbf{\Sigma}_1^{-1}\delta_1 \leq \frac{1}{\lambda_{1\min}}\delta_1^\top\delta_1$ and further derive minimum and maximum $\ell_2$ norm radii from Eq. 12:

$$\|\delta_1\|_{\text{lower}} = \sqrt{\lambda_{1\min}}\left(\Phi^{-1}\left(\underline{p}\right)\right), \tag{13}$$

$$\|\delta_1\|_{\text{upper}} = \sqrt{\lambda_{1\max}}\left(\Phi^{-1}\left(\underline{p}\right)\right), \tag{14}$$

where $\lambda_{1\min}$ and $\lambda_{1\max}$ are the minimum and maximum eigenvalue of $\mathbf{\Sigma}_1$. Inspired by Eiras et al. (2021), we can also use the ellipsoidal volume to measure the certified space. The volume of the ellipsoid is given by: $\mathcal{V}(\mathcal{R}) = r^m\sqrt{\pi^m}/\Gamma(m/2 + 1)\prod_{i=1}^m \xi_i$ (Kendall, 2004), which we use to obtain a proxy $\ell_2$ norm radius from Eq. 12:

$$\|\delta_1\|_{\text{volume}} = \left(\Phi^{-1}\left(\underline{p}\right)\right)\left(\sqrt{\pi}/\sqrt[m]{\Gamma(m/2 + 1)}\right)\sqrt[2m]{\prod_i^m \lambda_{1i}}, \tag{15}$$

where $\lambda_{1i}$ is the $i$-th eigenvalue of $\mathbf{\Sigma}_1$, and $m$ is the number of eigenvalues. In summary, the certified space of Eq. 12 can be regarded as a hyperellipsoid with three radii: $\|\delta_1\|_{\text{lower}}$ as the minor axis, $\|\delta_1\|_{\text{upper}}$ as the major axis, and $\|\delta_1\|_{\text{volume}}$ as a proxy radius of a hypersphere whose volume is proportional to the volume of this hyperellipsoid. Eq. 13, Eq. 14 and Eq. 15 are all quantifiable forms, Eq. 13 is the lower bound radius that guarantees robustness against the worst-case adversaries, Eq. 14 is the upper bound radius that indicates the maximum potential to resist adversaries, and Eq. 15 is the closest assessment to the certified space. Similarly, by setting $\delta_1$ as a zero matrix, we can obtain the three radii of $\delta_2$ ($\|\delta_2\|_{\text{lower}}$, $\|\delta_2\|_{\text{upper}}$, and $\|\delta_2\|_{\text{volume}}$) in the same manner. We can use these three radii of $\delta_1$ and $\delta_2$ to evaluate the probable robustness thoroughly.

## 5 EXPERIMENTS

This section describes the experimental settings, such as datasets, GM solvers, parameter settings, etc., and the evaluation criteria. Then, it compares the robustness certification and matching performance of CR-OSRS and RS-GM for six common GM solvers. Furthermore, it conducts ablation studies to elucidate the effect of different hyperparameters on the results.

## 5.1 EXPERIMENTS SETTINGS

In this section, we apply CR-OSRS and RS-GM to transform base solvers into smoothed ones with certified robustness for comparison and analysis. Note that the original RS is not suitable for

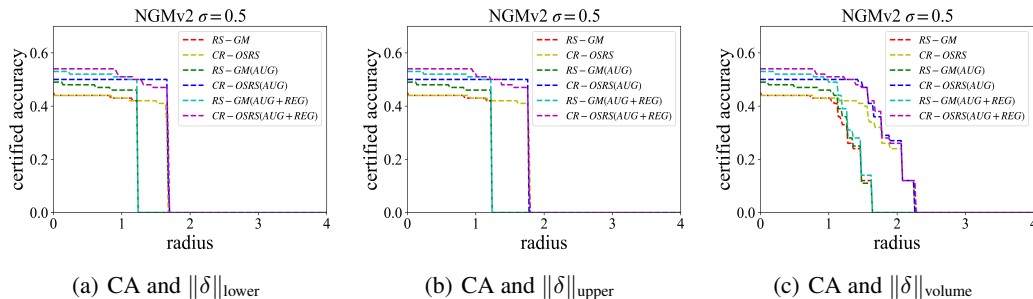

(a) CA and $\|\delta\|_{\text{lower}}$        (b) CA and $\|\delta\|_{\text{upper}}$        (c) CA and $\|\delta\|_{\text{volume}}$

Figure 2: CA achieved by CR-OSRS and RS-GM for NGMv2 on Pascal VOC when perturbing keypoint positions. "AUG" denotes data augmentation and "REG" denotes the regularization term in Eq. 11. Fig. 2 shows the result for original $\sigma = 0.5$, $s = 0.9$ in Eq. 5, $\beta = 0.01$ and $n = 2$ in Eq. 11.

solving robustness certification of functions with paired input, so we use a modified method RS-GM that follows Theorem 4.1, except replacing the smoothing distribution with an isotropic Gaussian distribution as in Cohen et al. (2019). Following the GM literature (Wang et al., 2021), we evaluate our method on the Pascal VOC dataset (Everingham et al., 2010) with Berkeley annotations (Bourdev & Malik, 2009), the Willow ObjectClass dataset (Cho et al., 2013) and SPair-71k dataset (Min et al., 2019) for six GM solvers, which are: GMN (Zanfir & Sminchisescu, 2018), PCA-GM (Wang et al., 2019), CIE-H (Yu et al., 2019a), NGMv2 (Wang et al., 2021), ASAR (Ren et al., 2022), COMMON (Lin et al., 2023). Unless otherwise specified, we use the same data processing and hyperparameter settings as in Wang et al. (2021). All the experiments are conducted on CPU (Intel(R) Core(TM) i7-7820X CPU @ 3.60GHz) and GPU (GTX 2080 Ti GPU).

## 5.2 EXPERIMENT RESULTS

This subsection reports the results on the Pascal VOC dataset under keypoint position perturbations. The results under image pixel perturbations as well as on the Willow ObjectClass dataset and SPair-71k dataset are presented in Appendix E.

**Robustness Certification Evaluation.** First, we use the sampling method presented in Sec. 4.4 to estimate the size of the certified space, where a larger space signifies stronger provable robustness. Specifically, we randomly generate 1,000 pairs of $(\delta_1, \delta_2)$ from a uniform distribution $\mathcal{U}(\sigma, \sigma)$. Then we insert the pairs into Eq. 9 and calculate the probability of pairs that satisfy Eq. 9. This probability for CR-OSRS with data augmentation is $83.5\%$ and is $40.7\%$ for RS-GM with data augmentation when $\sigma = 0.5$, $s = 0.9$ in Eq. 5, $\beta = 0.01$ and $n = 2$ in Eq. 11. This indicates that the certified space derived by CR-OSRS is larger than that derived by RS-GM, i.e., CR-OSRS achieves better robustness guarantees.

Second, to evaluate the three marginal radii ($\|\delta\|_{\text{lower}}$, $\|\delta\|_{\text{upper}}$, and $\|\delta\|_{\text{volume}}$) proposed in Sec. 4.4, we propose two indices: certified accuracy (CA) and average certified radius (ACR). Inspired by CA for classification (Cohen et al., 2019), we define CA for GM: $CA(R) = \mathbb{E}_{(x, \mathbf{X}_{gt})} \left[ \mathbb{I}(\|\delta_1\| \geq R) \mathbb{I}(\|\delta_2\| \geq R) \mathbb{I}(g_0(x) = 1) \mathbb{I}(\mathbf{X_c} = \mathbf{X}_{gt}) \right]$, where $\mathbb{I}$ is an indicator function, $\|\delta_1\|$ and $\|\delta_2\|$ denote the radii calculated by Eq. 13, Eq. 14, or Eq. 15, $R$ is the scale, $g_0$ represents the smoothed function defined in Eq. 7, $x$ denotes an element in the test set. Meanwhile, to measure the certified robustness of the entire test set, we refer to the ACR mentioned in Zhai et al. (2020) to propose the ACR for GM: $ACR = \mathbb{E}_{(x, \mathbf{X}_{gt})} \left[ \|\delta_1\| \|\delta_2\| \mathbb{I}(g_0(x) = 1) \mathbb{I}(\mathbf{X_c} = \mathbf{X}_{gt}) \right]$.

We examine the relationship of CA and three marginal radii for RS-GM and CR-OSRS in Fig. 2. We also compare the results of adding only data augmentation, as well as adding both the data augmentation and the regularization term, as in Eq. 11. The curve of CR-OSRS is almost always above RS-GM in Fig. 2, which implies greater certified robustness and matching accuracy. At the same time, it also demonstrates that the proposed data augmentation and the regularization term are effective. Note that there is no line for "REG" only in Fig. 2. This is because when there is no data augmentation, the outputs corresponding to all copy data described in Sec. 4.3 are the same, so the regularization term is always zero and the certification result is consistent with RS-GM or CR-OSRS. To measure the overall provable robustness of the entire dataset, we calculate the ACR of CR-OSRS and RS-GM for six GM solvers in Table 1. It is obvious that the overall provable robustness of CR-OSRS is better than that of RS-GM on different GM solvers. We observe a positive association between the performance of the base and the smoothed models. Namely, the smoothed model exhibits higher certified robustness and matching performance as the base model's performance increases.

Table 1: ACR achieved by CR-OSRS and RS-GM for six GM solvers on Pascal VOC under keypoint position perturbations. Table 1 shows the result for $\sigma = 0.5$, $s = 0.9$, $\beta = 0.01$ and $n = 2$.

| | CR-OSRS+AUG+REG | | | RS-GM+AUG+REG | | |
|---|---|---|---|---|---|---|
| | $\|\delta\|_{\mathbf{lower}}$ | $\|\delta\|_{\mathbf{upper}}$ | $\|\delta\|_{\mathbf{volume}}$ | $\|\delta\|_{\mathbf{lower}}$ | $\|\delta\|_{\mathbf{upper}}$ | $\|\delta\|_{\mathbf{volume}}$ |
| **COMMON (Lin et al., 2023)** | 1.550 | 1.751 | 1.900 | 0.952 | 0.952 | 1.069 |
| **ASAR (Ren et al., 2022)** | 1.541 | 1.648 | 1.968 | 0.683 | 0.683 | 0.841 |
| **NGMv2 (Wang et al., 2021)** | 1.425 | 1.586 | 1.934 | 0.778 | 0.778 | 1.010 |
| **CIE-H (Yu et al., 2019a)** | 0.987 | 1.167 | 1.354 | 0.572 | 0.572 | 0.731 |
| **PCA-GM (Wang et al., 2019)** | 0.954 | 1.158 | 1.340 | 0.546 | 0.546 | 0.686 |
| **GMN (Zanfir & Sminchisescu, 2018)** | 0.899 | 1.076 | 1.253 | 0.514 | 0.514 | 0.617 |

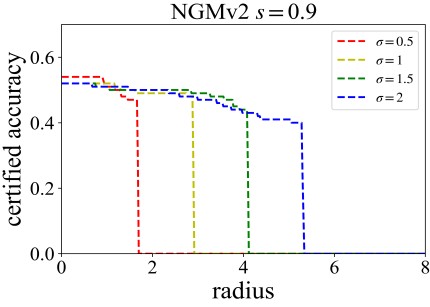

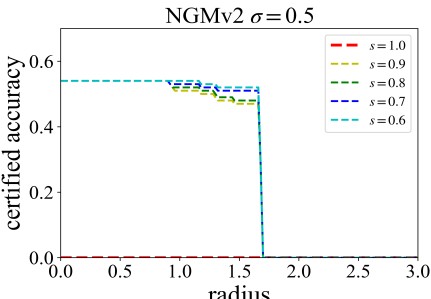

(a) CA and $\|\delta\|_{\text{lower}}$ when varying original $\sigma$. $\sigma$ determines the scale of $\mathbf{\Sigma_1}$ and $\mathbf{\Sigma_2}$ that controls the trade-off between certified robustness and certified accuracy.

(b) CA and $\|\delta\|_{\text{lower}}$ when varying the original $s$. Reducing $s$ enhances the certified robustness of the model, as it enlarges the output subspace in Eq. 5 and relaxes the constraints on the output.

Figure 3: Projections for the certified accuracy if the original $\sigma$ and similarity threshold $s$ had been larger or smaller. Fig. 3 shows the result for CR-OSRS trained by data augmentation and regularization term with $\beta = 0.01$ and $n = 2$ for NGMv2 on Pascal VOC.

**Hyperparameter Analysis.** Our method introduces the following hyperparameters: original $\sigma$, similarity threshold $s$ for subspace construction as defined in Eq. 5, the constraint hyperparameter $\kappa$, number of copies $n$ and regularization hyperparameter $\beta$ as shown in Eq. 11 as well as $k$ for Monte Carlo sampling. This subsection examines the effect of $\sigma$ and $s$, and refers the readers to Appendix E for the other hyperparameters. $\sigma$ is varied from $\sigma \in \{0.5, 1.0, 1.5, 2.0\}$ and the certified accuracy with each $\sigma$ is plotted in Fig. 3(a). Generally, a lower $\sigma$ results in higher certified accuracy and lower certified radii, while a higher $\sigma$ allows for larger certified radii but lower certified accuracy. $s$ is varied from $s \in \{0.6, 0.7, 0.8, 0.9, 1.0\}$ and the certified accuracy achieved by CR-OSRS with each $s$ is plotted in Fig. 3(b). When $s = 1$, the subspace in Eq. 5 degenerates into a single matrix, which implies a stringent robustness guarantee that the output remains invariant under any perturbation. However, as shown in Fig. 3(b), when $s = 1$, the accuracy is always zero, which is consistent with the discussion in Sec. 4.1. The certification may fail or yield a small certification range due to the absence of a dominant matrix.

## 6 CONCLUSION AND OUTLOOK

This paper introduces the first definition of certified robustness for visual graph matching and proposes a novel method, named CR-OSRS. This method uses the correlation between keypoints to construct a joint smoothing distribution and devises a global optimization algorithm to determine the optimal smoothing range that balances provable robustness and matching performance. Furthermore, it presents a data augmentation technique based on the joint Gaussian distribution and a regularization term based on output similarity to improve model performance during the training phase. Then it derives an $\ell_2$-norm certified space and suggests two quantitative methods (sampling and marginal radii) to address the challenge of quantifying the certified space. Finally, it conducts experiments on different GM solvers and datasets and achieves state-of-the-art robustness certification.

**Potential impact & limitations.** A significant direction is to enable robustness certification on combinatorial solvers whereby GM is one of such cases. We expect that this work can inspire subsequent research in this promising area where theoretical results are welcomed given recent intensive empirical studies, e.g., Bengio et al. (2021); Yan et al. (2020).

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

## A PROOFS

In this section, we present the full proofs for Theorem. 4.1. The main tool for our proofs is the Neyman-Pearson lemma for two variables, which we establish in Appendix A.1. Based on this lemma, we obtain the certified result in Appendix A.2. Finally, we provide the details of the linear transformation used for certification in Appendix A.3.

### A.1 NEYMAN-PEARSON FOR TWO VARIABLES

**Lemma A.1** (Neyman-Pearson for two variables). *Let $X_1$ and $Y_1$ be random variables in $\mathbb{R}^d$ with densities $\mu_{X_1}$ and $\mu_{Y_1}$. Then, let $X_2$ and $Y_2$ be random variables in $\mathbb{R}^d$ with densities $\mu_{X_2}$ and $\mu_{Y_2}$. Let $h : \mathbb{R}^d \times \mathbb{R}^d \to \{0, 1\}$ be any deterministic or random function with an input pair. Then:*

*1. If $S_1 \times S_2 = \left\{ z_1 \in \mathbb{R}^d, z_2 \in \mathbb{R}^d : \frac{\mu_{Y_1}(z_1)\mu_{Y_2}(z_2)}{\mu_{X_1}(z_1)\mu_{X_2}(z_2)} \leq t \right\}$ for some $t > 0$ and $P(h(X_1, X_2) = 1) \geq P((X_1, X_2) \in S_1 \times S_2)$, then $P(h(Y_1, Y_2) = 1) \geq P((Y_1, Y_2) \in S_1 \times S_2)$.*

*2. If $S_1 \times S_2 = \left\{ z_1 \in \mathbb{R}^d, z_2 \in \mathbb{R}^d : \frac{\mu_{Y_1}(z_1)\mu_{Y_2}(z_2)}{\mu_{X_1}(z_1)\mu_{X_2}(z_2)} \geq t \right\}$ for some $t > 0$ and $P(h(X_1, X_2) = 1) \leq P((X_1, X_2) \in S_1 \times S_2)$, then $P(h(Y_1, Y_2) = 1) \leq P((Y_1, Y_2) \in S_1 \times S_2)$.*

*Proof.* We denote the complement of $S_1 \times S_2$ as $S^c$.

$$P(h(Y_1, Y_2) = 1) - P((Y_1, Y_2) \in S_1 \times S_2)$$

$$= \int_{\mathbb{R}^d} \int_{\mathbb{R}^d} h(1 \mid z_1, z_2)\mu_{Y_1}(z_1)\mu_{Y_2}(z_2)dz_1dz_2 - \int\int_{S_1 \times S_2} \mu_{Y_1}(z_1)\mu_{Y_2}(z_2)dz_1dz_2$$

$$= \left[ \int\int_{S_1 \times S_2} h(1 \mid z_1, z_2)\mu_{Y_1}(z_1)\mu_{Y_2}(z_2)dz_1dz_2 + \int\int_{S^c} h(1 \mid z_1, z_2)\mu_{Y_1}(z_1)\mu_{Y_2}(z_2)dz_1dz_2 \right]$$

$$- \left[ \int\int_{S_1 \times S_2} h(1 \mid z_1, z_2)\mu_{Y_1}(z_1)\mu_{Y_2}(z_2)dz_1dz_2 + \int\int_{S_1 \times S_2} h(0 \mid z_1, z_2)\mu_{Y_1}(z_1)\mu_{Y_2}(z_2)dz_1dz_2 \right]$$

$$= \int\int_{S^c} h(1 \mid z_1, z_2)\mu_{Y_1}(z_1)\mu_{Y_2}(z_2)dz_1dz_2 - \int\int_{S_1 \times S_2} h(0 \mid z_1, z_2)\mu_{Y_1}(z_1)\mu_{Y_2}(z_2)dz_1dz_2$$

$$\geq t \left[ \int\int_{S^c} h(1 \mid z_1, z_2)\mu_{X_1}(z_1)\mu_{X_2}(z_2)dz_1dz_2 - \int\int_{S_1 \times S_2} h(0 \mid z_1, z_2)\mu_{X_1}(z_1)\mu_{X_2}(z_2)dz_1dz_2 \right]$$

$$= t[\int\int_{S^c} h(1 \mid z_1, z_2)\mu_{X_1}(z_1)\mu_{X_2}(z_2)dz_1dz_2 + \int\int_{S_1 \times S_2} h(1 \mid z_1, z_2)\mu_{X_1}(z_1)\mu_{X_2}(z_2)dz_1dz_2$$

$$- \int\int_{S_1 \times S_2} h(1 \mid z_1, z_2)\mu_{X_1}(z_1)\mu_{X_2}(z_2)dz_1dz_2 - \int\int_{S_1 \times S_2} h(0 \mid z_1, z_2)\mu_{X_1}(z_1)\mu_{X_2}(z_2)dz_1dz_2]$$

$$= t \left[ \int_{\mathbb{R}^d} \int_{\mathbb{R}^d} h(1 \mid z_1, z_2)\mu_{X_1}(z_1)\mu_{X_2}(z_2)dz_1dz_2 - \int\int_{S_1 \times S_2} \mu_{X_1}(z_1)\mu_{X_2}(z_2)dz_1dz_2 \right]$$

$$= t[P(h(X_1, X_2) = 1) - P((X_1, X_2) \in S_1 \times S_2)]$$

$$\geq 0$$

$$\tag{16}$$

$\square$

Next, we prove Lemma A.2, which is a special case of Lemma A.1 and states the Neyman-Pearson lemma for two joint Gaussian noise variables.

**Lemma A.2** (Neyman-Pearson for Two Joint Gaussian Noise). *Let $X_1 \sim \mathcal{N}(x_1, \mathbf{\Sigma_1})$, $X_2 \sim \mathcal{N}(x_2, \mathbf{\Sigma_2})$ and $Y_1 \sim \mathcal{N}(x_1 + \delta_1, \mathbf{\Sigma_1})$, $Y_2 \sim \mathcal{N}(x_2 + \delta_2, \mathbf{\Sigma_2})$. Let $h : \mathbb{R}^d \times \mathbb{R}^d \to \{0, 1\}$ be any deterministic or random function. Then:*

*1. If $S_1 \times S_2 = \left\{ z_1 \in \mathbb{R}^d, z_2 \in \mathbb{R}^d : \delta_1^T \mathbf{\Sigma_1}^{-1} z_1 + \delta_2^T \mathbf{\Sigma_2}^{-1} z_2 \leq \beta \right\}$ for some $\beta$ and $P(h(X_1, X_2) = 1) \geq P((X_1, X_2) \in S_1 \times S_2)$, then $P(h(Y_1, Y_2) = 1) \geq P((Y_1, Y_2) \in S_1 \times S_2)$.*

*2. If $S_1 \times S_2 = \left\{ z_1 \in \mathbb{R}^d, z_2 \in \mathbb{R}^d : \delta_1^T \mathbf{\Sigma_1}^{-1} z_1 + \delta_2^T \mathbf{\Sigma_2}^{-1} z_2 \geq \beta \right\}$ for some $\beta$ and $P(h(X_1, X_2) = 1) \leq P((X_1, X_2) \in S_1 \times S_2)$, then $P(h(Y_1, Y_2) = 1) \leq P((Y_1, Y_2) \in S_1 \times S_2)$.*

*Proof.* This lemma is the special case of Neyman-Pearson for two variables when $X_1$, $X_2$, $Y_1$, and $Y_2$ are joint Gaussian noises. It suffices to simply show that for any $\beta$, there is some $t > 0$ for which:

$$
\begin{aligned}
\left\{z_1, z_2 : \delta_1^T \boldsymbol{\Sigma_1}^{-1} z_1 + \delta_2^T \boldsymbol{\Sigma_2}^{-1} z_2 \le \beta\right\} &= \left\{z_1, z_2 : \frac{\mu_{Y_1}(z_1)\mu_{Y_2}(z_2)}{\mu_{X_1}(z_1)\mu_{X_2}(z_2)} \le t\right\}, \\
\left\{z_1, z_2 : \delta_1^T \boldsymbol{\Sigma_1}^{-1} z_1 + \delta_2^T \boldsymbol{\Sigma_2}^{-1} z_2 \ge \beta\right\} &= \left\{z_1, z_2 : \frac{\mu_{Y_1}(z_1)\mu_{Y_2}(z_2)}{\mu_{X_1}(z_1)\mu_{X_2}(z_2)} \ge t\right\}.
\end{aligned}
\tag{17}
$$

For ease of representation, we use $M_1 \in \mathbb{R}^{d \times d}$ (with element $m_{1_{ij}}$) instead of $\boldsymbol{\Sigma_1}^{-1}$ and $M_2 \in \mathbb{R}^{d \times d}$ (with element $m_{2_{ij}}$) instead of $\boldsymbol{\Sigma_2}^{-1}$. The likelihood ratio for this choice of $X_1$, $X_2$, $Y_1$ and $Y_2$ turns out to be:

$$
\begin{aligned}
&\frac{\mu_{Y_1}(z_1)\mu_{Y_2}(z_2)}{\mu_{X_1}(z_1)\mu_{X_2}(z_2)} \\
&= \frac{\exp\left(-\frac{1}{2}(z_1 - (x_1 + \delta_1))^T \boldsymbol{\Sigma_1}^{-1}(z_1 - (x_1 + \delta_1))\right)}{\exp\left(-\frac{1}{2}(z_1 - x_1)^T \boldsymbol{\Sigma_1}^{-1}(z_1 - x_1)\right)} \times \frac{\exp\left(-\frac{1}{2}(z_2 - (x_2 + \delta_2))^T \boldsymbol{\Sigma_2}^{-1}(z_2 - (x_2 + \delta_2))\right)}{\exp\left(-\frac{1}{2}(z_2 - x_2)^T \boldsymbol{\Sigma_2}^{-1}(z_2 - x_2)\right)} \\
&= \frac{\exp\left(-\frac{1}{2}\sum_i^d \sum_j^d (z_{1_i} - (x_{1_i} + \delta_{1_i})) m_{1_{ij}} \left(z_{1_j} - \left(x_{1_j} + \delta_{1_j}\right)\right)\right)}{\exp\left(-\frac{1}{2}\sum_i^d \sum_j^d (z_{1_i} - x_{1_i}) m_{1_{ij}} \left(z_{1_j} - x_{1_j}\right)\right)} \\
&\quad \times \frac{\exp\left(-\frac{1}{2}\sum_i^d \sum_j^d (z_{2_i} - (x_{2_i} + \delta_{2_i})) m_{2_{ij}} \left(z_{2_j} - \left(x_{2_j} + \delta_{2_j}\right)\right)\right)}{\exp\left(-\frac{1}{2}\sum_i^d \sum_j^d (z_{2_i} - x_{2_i}) m_{2_{ij}} \left(z_{2_j} - x_{2_j}\right)\right)} \\
&= \exp\left(\delta_1^T \boldsymbol{\Sigma_1}^{-1} z_1 - \delta_1^T \boldsymbol{\Sigma_1}^{-1} x_1 - \frac{1}{2}\delta_1^T \boldsymbol{\Sigma_1}^{-1}\delta_1\right) \times \exp\left(\delta_2^T \boldsymbol{\Sigma_2}^{-1} z_2 - \delta_2^T \boldsymbol{\Sigma_2}^{-1} x_2 - \frac{1}{2}\delta_2^T \boldsymbol{\Sigma_2}^{-1}\delta_2\right) \\
&= \exp\left(\delta_1^T \boldsymbol{\Sigma_1}^{-1} z_1 + \delta_2^T \boldsymbol{\Sigma_2}^{-1} z_2 - \delta_1^T \boldsymbol{\Sigma_1}^{-1} x_1 - \frac{1}{2}\delta_1^T \boldsymbol{\Sigma_1}^{-1}\delta_1 - \delta_2^T \boldsymbol{\Sigma_2}^{-1} x_2 - \frac{1}{2}\delta_2^T \boldsymbol{\Sigma_2}^{-1}\delta_2\right) \\
&= \exp\left(\delta_1^T \boldsymbol{\Sigma_1}^{-1} z_1 + \delta_2^T \boldsymbol{\Sigma_2}^{-1} z_2 + b\right) \le t,
\end{aligned}
$$

where $b$ is a constant, specifically $b = -\delta_1^T \boldsymbol{\Sigma_1}^{-1} x_1 - \frac{1}{2}\delta_1^T \boldsymbol{\Sigma_1}^{-1}\delta_1 - \delta_2^T \boldsymbol{\Sigma_2}^{-1} x_2 - \frac{1}{2}\delta_2^T \boldsymbol{\Sigma_2}^{-1}\delta_2$. Therefore given any $\beta$, we may take $t = \exp(\beta + b)$ and obtain this correlation:

$$
\begin{aligned}
\delta_1^T \boldsymbol{\Sigma_1}^{-1} z_1 + \delta_2^T \boldsymbol{\Sigma_2}^{-1} z_2 \le \beta &\iff \exp(\beta + b) \le t, \\
\delta_1^T \boldsymbol{\Sigma_1}^{-1} z_1 + \delta_2^T \boldsymbol{\Sigma_2}^{-1} z_2 \ge \beta &\iff \exp(\beta + b) \ge t.
\end{aligned}
\tag{18}
$$

$\square$

### A.2 PROOF OF THE CERTIFIED ROBUSTNESS

This subsection presents the logic for proving robustness guarantees and derives the certified spaces for these guarantees in Eq. 9.

To show that $g_0\left(\mathbf{c}^1, \mathbf{c}^2, \mathbf{z}^1 + \delta_1, \mathbf{z}^2 + \delta_2\right) = 1$, it follows from the definition of $g_0$ that we need to show that:

$$
P(f\left(\mathbf{c}^1, \mathbf{c}^2, \mathbf{z}^1 + \varepsilon_1 + \delta_1, \mathbf{z}^2 + \varepsilon_2 + \delta_2\right) \in \mathcal{X}') \ge P(f\left(\mathbf{c}^1, \mathbf{c}^2, \mathbf{z}^1 + \varepsilon_1 + \delta_1, \mathbf{z}^2 + \varepsilon_2 + \delta_2\right) \notin \mathcal{X}').
$$

We define two random variables:

$$
\begin{aligned}
I &:= \left(\mathbf{c}^1, \mathbf{c}^2, \mathbf{z}^1 + \varepsilon_1, \mathbf{z}^2 + \varepsilon_2\right) = \left(\mathbf{c}^1, \mathbf{c}^2, \mathcal{N}\left(\mathbf{z}^1, \boldsymbol{\Sigma_1}\right), \mathcal{N}\left(\mathbf{z}^2, \boldsymbol{\Sigma_2}\right)\right) \\
O &:= \left(\mathbf{c}^1, \mathbf{c}^2, \mathbf{z}^1 + \varepsilon_1 + \delta_1, \mathbf{z}^2 + \varepsilon_2 + \delta_2\right) = \left(\mathbf{c}^1, \mathbf{c}^2, \mathcal{N}\left(\mathbf{z}^1 + \delta_1, \boldsymbol{\Sigma_1}\right), \mathcal{N}\left(\mathbf{z}^2 + \delta_2, \boldsymbol{\Sigma_2}\right)\right).
\end{aligned}
$$

We know that:

$$
P(f(I) \in \mathcal{X}') \ge \underline{p}.
\tag{19}
$$

Our goal is to show that

$$
P(f(O) \in \mathcal{X}') > P(f(O) \notin \mathcal{X}').
\tag{20}
$$

According to lemma A.2, we can define the half-spaces:

$$A = \left\{ z_1, z_2 : \delta_1^T \mathbf{\Sigma_1}^{-1}(z_1 - \mathbf{z}^1) + \delta_2^T \mathbf{\Sigma_2}^{-1}(z_2 - \mathbf{z}^2) \leq \|\delta_1^T \mathbf{\Sigma_1}^{-1}\mathbf{B_1} + \delta_2^T \mathbf{\Sigma_2}^{-1}\mathbf{B_2}\| \Phi^{-1}\left(\underline{p}\right) \right\},$$

$$B = \left\{ z_1, z_2 : \delta_1^T \mathbf{\Sigma_1}^{-1}(z_1 - \mathbf{z}^1) + \delta_2^T \mathbf{\Sigma_2}^{-1}(z_2 - \mathbf{z}^2) \geq \|\delta_1^T \mathbf{\Sigma_1}^{-1}\mathbf{B_1} + \delta_2^T \mathbf{\Sigma_2}^{-1}\mathbf{B_2}\| \Phi^{-1}\left(\underline{p}\right) \right\}.$$

Claim 1 shows that $P(I \in A) = \underline{p}$, therefore we can obtain $P(f(I) \in \mathcal{X}') \geq P(I \in A)$. Hence we may apply Lemma A.2 to conclude:

$$P(f(O) \in \mathcal{X}') \geq P(O \in A). \tag{21}$$

Similarly, we obtain $P(f(I) \notin \mathcal{X}') \leq P(I \in B)$. Hence we may apply Lemma A.2 to conclude:

$$P(f(O) \notin \mathcal{X}') \leq P(O \in B). \tag{22}$$

Combining Eq. 21 and 22, we can obtain the conditions of Eq. 20:

$$P(f(O) \in \mathcal{X}') \geq P(O \in A) > P(O \in B) \geq P(f(O) \notin \mathcal{X}'). \tag{23}$$

According to Claim 3 and Claim 4, we can obtain $P(O \in A)$ and $P(O \in B)$ as:

$$
\begin{aligned}
P(O \in A) &= \Phi \left( \Phi^{-1}\left(\underline{p}\right) - \frac{\delta_1^T \mathbf{\Sigma_1}^{-1}\delta_1 + \delta_2^T \mathbf{\Sigma_2}^{-1}\delta_2}{\|\delta_1^T \mathbf{\Sigma_1}^{-1}\mathbf{B_1} + \delta_2^T \mathbf{\Sigma_2}^{-1}\mathbf{B_2}\|} \right), \\
P(O \in B) &= \Phi \left( -\Phi^{-1}\left(\underline{p}\right) + \frac{\delta_1^T \mathbf{\Sigma_1}^{-1}\delta_1 + \delta_2^T \mathbf{\Sigma_2}^{-1}\delta_2}{\|\delta_1^T \mathbf{\Sigma_1}^{-1}\mathbf{B_1} + \delta_2^T \mathbf{\Sigma_2}^{-1}\mathbf{B_2}\|} \right).
\end{aligned} \tag{24}
$$

Finally, we obtain that $P(O \in A) > P(O \in B)$ if and only if:

$$\frac{\delta_1^T \mathbf{\Sigma_1}^{-1}\delta_1 + \delta_2^T \mathbf{\Sigma_2}^{-1}\delta_2}{\|\delta_1^T \mathbf{\Sigma_1}^{-1}\mathbf{B_1} + \delta_2^T \mathbf{\Sigma_2}^{-1}\mathbf{B_2}\|} < \Phi^{-1}\left(\underline{p}\right)).$$

### A.3 LINEAR TRANSFORMATION AND DERIVATION

This subsection begins with Lemma A.3, which is the main tool for deriving all claims. Then, we present the proof process of claims, which is applied in Sec. A.2.

**Lemma A.3** (Joint Gaussian Distribution). *If there is a random matrix $X \sim \mathcal{N}(\mu, \mathbf{\Sigma})$, where $\mu \in \mathbb{R}^n$ is the mean matrix. A positive semi-definite real symmetric matrix $\mathbf{\Sigma} \in \mathbb{S}_{++}^{n \times n}$ is the covariance matrix of $X$. There is a full rank matrix $\mathbf{B} \in \mathbb{R}^{n \times n}$, which makes $X = \mathbf{B}Z + \mu$, $Z \sim \mathcal{N}(\mathbf{0}, I)$ and $\mathbf{B}^\top \mathbf{B} = \Sigma$.*

We obtain four claims based on linear transformation:

**Claim 1.** $P(I \in A) = \underline{p}$

*Proof.* Recall that $A = \left\{ z_1, z_2 : \delta_1^T \mathbf{\Sigma_1}^{-1}(z_1 - \mathbf{z}^1) + \delta_2^T \mathbf{\Sigma_2}^{-1}(z_2 - \mathbf{z}^2) \leq \|\delta_1^T \mathbf{\Sigma_1}^{-1}\mathbf{B_1} + \delta_2^T \mathbf{\Sigma_2}^{-1}\mathbf{B_2}\| \Phi^{-1}\left(\underline{p}\right) \right\}$, according to lemma A.3, we can obtain:

$$
\begin{aligned}
P(I \in A) &= P\left( \delta_1^T \mathbf{\Sigma_1}^{-1}(\mathcal{N}\left(\mathbf{z}^1, \mathbf{\Sigma_1}\right) - \mathbf{z}^1) + \delta_2^T \mathbf{\Sigma_2}^{-1}(\mathcal{N}\left(\mathbf{z}^2, \mathbf{\Sigma_2}\right) - \mathbf{z}^2) \leq \|\delta_1^T \mathbf{\Sigma_1}^{-1}\mathbf{B_1} + \delta_2^T \mathbf{\Sigma_2}^{-1}\mathbf{B_2}\| \Phi^{-1}\left(\underline{p}\right) \right) \\
&= P\left( \delta_1^T \mathbf{\Sigma_1}^{-1}\mathcal{N}(0, \mathbf{\Sigma_1}) + \delta_2^T \mathbf{\Sigma_2}^{-1}\mathcal{N}(0, \mathbf{\Sigma_2}) \leq \|\delta_1^T \mathbf{\Sigma_1}^{-1}\mathbf{B_1} + \delta_2^T \mathbf{\Sigma_2}^{-1}\mathbf{B_2}\| \Phi^{-1}\left(\underline{p}\right) \right) \\
&= P\left( \delta_1^T \mathbf{\Sigma_1}^{-1}\mathbf{B_1}\mathcal{N}(0, I) + \delta_2^T \mathbf{\Sigma_2}^{-1}\mathbf{B_2}\mathcal{N}(0, I) \leq \|\delta_1^T \mathbf{\Sigma_1}^{-1}\mathbf{B_1} + \delta_2^T \mathbf{\Sigma_2}^{-1}\mathbf{B_2}\| \Phi^{-1}\left(\underline{p}\right) \right) \\
&= P\left( \|\delta_1^T \mathbf{\Sigma_1}^{-1}\mathbf{B_1} + \delta_2^T \mathbf{\Sigma_2}^{-1}\mathbf{B_2}\|\mathcal{N}(0, 1) \leq \|\delta_1^T \mathbf{\Sigma_1}^{-1}\mathbf{B_1} + \delta_2^T \mathbf{\Sigma_2}^{-1}\mathbf{B_2}\| \Phi^{-1}\left(\underline{p}\right) \right) \\
&= \Phi\left( \Phi^{-1}\left(\underline{p}\right) \right) \\
&= \underline{p}.
\end{aligned}
$$

$\square$

**Claim 2.** $P(I \in B) = 1 - \underline{p}$

*Proof.* Recall that $B = \left\{ z_1, z_2 : \delta_1^T \mathbf{\Sigma_1}^{-1}(z_1 - \mathbf{z}^1) + \delta_2^T \mathbf{\Sigma_2}^{-1}(z_2 - \mathbf{z}^2) \geq \|\delta_1^T \mathbf{\Sigma_1}^{-1} \mathbf{B_1} + \delta_2^T \mathbf{\Sigma_2}^{-1} \mathbf{B_2}\| \Phi^{-1}\left(\underline{p}\right) \right\}$, according to lemma A.3, we can obtain:

$$
\begin{aligned}
P(I \in A) &= P\left( \delta_1^T \mathbf{\Sigma_1}^{-1}(\mathcal{N}\left(\mathbf{z}^1, \mathbf{\Sigma_1}\right) - \mathbf{z}^1) + \delta_2^T \mathbf{\Sigma_2}^{-1}(\mathcal{N}\left(\mathbf{z}^2, \mathbf{\Sigma_2}\right) - \mathbf{z}^2) \geq \|\delta_1^T \mathbf{\Sigma_1}^{-1} \mathbf{B_1} + \delta_2^T \mathbf{\Sigma_2}^{-1} \mathbf{B_2}\| \Phi^{-1}\left(\underline{p}\right) \right) \\
&= P\left( \delta_1^T \mathbf{\Sigma_1}^{-1} \mathcal{N}\left(0, \mathbf{\Sigma_1}\right) + \delta_2^T \mathbf{\Sigma_2}^{-1} \mathcal{N}\left(0, \mathbf{\Sigma_2}\right) \geq \|\delta_1^T \mathbf{\Sigma_1}^{-1} \mathbf{B_1} + \delta_2^T \mathbf{\Sigma_2}^{-1} \mathbf{B_2}\| \Phi^{-1}\left(\underline{p}\right) \right) \\
&= P\left( \delta_1^T \mathbf{\Sigma_1}^{-1} \mathbf{B_1} \mathcal{N}\left(0, I\right) + \delta_2^T \mathbf{\Sigma_2}^{-1} \mathbf{B_2} \mathcal{N}\left(0, I\right) \geq \|\delta_1^T \mathbf{\Sigma_1}^{-1} \mathbf{B_1} + \delta_2^T \mathbf{\Sigma_2}^{-1} \mathbf{B_2}\| \Phi^{-1}\left(\underline{p}\right) \right) \\
&= P\left( \|\delta_1^T \mathbf{\Sigma_1}^{-1} \mathbf{B_1} + \delta_2^T \mathbf{\Sigma_2}^{-1} \mathbf{B_2}\| \mathcal{N}\left(0, 1\right) \geq \|\delta_1^T \mathbf{\Sigma_1}^{-1} \mathbf{B_1} + \delta_2^T \mathbf{\Sigma_2}^{-1} \mathbf{B_2}\| \Phi^{-1}\left(\underline{p}\right) \right) \\
&= 1 - \Phi\left( \Phi^{-1}\left(\underline{p}\right) \right) \\
&= 1 - \underline{p}.
\end{aligned}
$$

$\square$

**Claim 3.** $P(O \in A) = \Phi\left( \Phi^{-1}\left(\underline{p}\right) - \frac{\delta_1^T \mathbf{\Sigma_1}^{-1} \delta_1 + \delta_2^T \mathbf{\Sigma_2}^{-1} \delta_2}{\|\delta_1^T \mathbf{\Sigma_1}^{-1} \mathbf{B_1} + \delta_2^T \mathbf{\Sigma_2}^{-1} \mathbf{B_2}\|} \right)$

*Proof.* Recall that $A = \left\{ z_1, z_2 : \delta_1^T \mathbf{\Sigma_1}^{-1}(z_1 - \mathbf{z}^1) + \delta_2^T \mathbf{\Sigma_2}^{-1}(z_2 - \mathbf{z}^2) \leq \|\delta_1^T \mathbf{\Sigma_1}^{-1} \mathbf{B_1} + \delta_2^T \mathbf{\Sigma_2}^{-1} \mathbf{B_2}\| \Phi^{-1}\left(\underline{p}\right) \right\}$ and $O \sim \left( \mathbf{c}^1, \mathbf{c}^2, \mathcal{N}\left(\mathbf{z}^1 + \delta_1, \mathbf{\Sigma_1}\right), \mathcal{N}\left(\mathbf{z}^2 + \delta_2, \mathbf{\Sigma_2}\right) \right)$, according to lemma A.3, we can obtain:

$$
\begin{aligned}
&P(O \in A) \\
&= P\left( \delta_1^T \mathbf{\Sigma_1}^{-1}(\mathcal{N}\left(\mathbf{z}^1 + \delta_1, \mathbf{\Sigma_1}\right) - \mathbf{z}^1) + \delta_2^T \mathbf{\Sigma_2}^{-1}(\mathcal{N}\left(\mathbf{z}^2 + \delta_2, \mathbf{\Sigma_2}\right) - \mathbf{z}^2) \leq \|\delta_1^T \mathbf{\Sigma_1}^{-1} \mathbf{B_1} + \delta_2^T \mathbf{\Sigma_2}^{-1} \mathbf{B_2}\| \Phi^{-1}\left(\underline{p}\right) \right) \\
&= P\left( \delta_1^T \mathbf{\Sigma_1}^{-1} \mathcal{N}\left(\delta_1, \mathbf{\Sigma_1}\right) + \delta_2^T \mathbf{\Sigma_2}^{-1} \mathcal{N}\left(\delta_2, \mathbf{\Sigma_2}\right) \leq \|\delta_1^T \mathbf{\Sigma_1}^{-1} \mathbf{B_1} + \delta_2^T \mathbf{\Sigma_2}^{-1} \mathbf{B_2}\| \Phi^{-1}\left(\underline{p}\right) \right) \\
&= P\left( \delta_1^T \mathbf{\Sigma_1}^{-1}(\mathbf{B_1} \mathcal{N}\left(0, I\right) + \delta_1) + \delta_2^T \mathbf{\Sigma_2}^{-1}(\mathbf{B_2} \mathcal{N}\left(0, I\right) + \delta_2) \leq \|\delta_1^T \mathbf{\Sigma_1}^{-1} \mathbf{B_1} + \delta_2^T \mathbf{\Sigma_2}^{-1} \mathbf{B_2}\| \Phi^{-1}\left(\underline{p}\right) \right) \\
&= P\left( \|\delta_1^T \mathbf{\Sigma_1}^{-1} \mathbf{B_1} + \delta_2^T \mathbf{\Sigma_2}^{-1} \mathbf{B_2}\| \mathcal{N}\left(0, 1\right) + \delta_1^T \mathbf{\Sigma_1}^{-1} \delta_1 + \delta_2^T \mathbf{\Sigma_2}^{-1} \delta_2 \leq \|\delta_1^T \mathbf{\Sigma_1}^{-1} \mathbf{B_1} + \delta_2^T \mathbf{\Sigma_2}^{-1} \mathbf{B_2}\| \Phi^{-1}\left(\underline{p}\right) \right) \\
&= P\left( \mathcal{N}\left(0, 1\right) \leq \Phi^{-1}\left(\underline{p}\right) - \frac{\delta_1^T \mathbf{\Sigma_1}^{-1} \delta_1 + \delta_2^T \mathbf{\Sigma_2}^{-1} \delta_2}{\|\delta_1^T \mathbf{\Sigma_1}^{-1} \mathbf{B_1} + \delta_2^T \mathbf{\Sigma_2}^{-1} \mathbf{B_2}\|} \right) \\
&= \Phi\left( \Phi^{-1}\left(\underline{p}\right) - \frac{\delta_1^T \mathbf{\Sigma_1}^{-1} \delta_1 + \delta_2^T \mathbf{\Sigma_2}^{-1} \delta_2}{\|\delta_1^T \mathbf{\Sigma_1}^{-1} \mathbf{B_1} + \delta_2^T \mathbf{\Sigma_2}^{-1} \mathbf{B_2}\|} \right).
\end{aligned}
$$

$\square$

**Claim 4.** $P(O \in \mathbf{B}) = \Phi\left( -\Phi^{-1}\left(\underline{p}\right) + \frac{\delta_1^T \mathbf{\Sigma_1}^{-1} \delta_1 + \delta_2^T \mathbf{\Sigma_2}^{-1} \delta_2}{\|\delta_1^T \mathbf{\Sigma_1}^{-1} \mathbf{B_1} + \delta_2^T \mathbf{\Sigma_2}^{-1} \mathbf{B_2}\|} \right)$

*Proof.* Recall that $B = \left\{ z_1, z_2 : \delta_1^T \mathbf{\Sigma_1}^{-1}(z_1 - \mathbf{z}^1) + \delta_2^T \mathbf{\Sigma_2}^{-1}(z_2 - \mathbf{z}^2) \geq \|\delta_1^T \mathbf{\Sigma_1}^{-1} \mathbf{B_1} + \delta_2^T \mathbf{\Sigma_2}^{-1} \mathbf{B_2}\| \Phi^{-1}\left(\underline{p}\right) \right\}$ and $O \sim \left( \mathbf{c}^1, \mathbf{c}^2, \mathcal{N}\left(\mathbf{z}^1 + \delta_1, \mathbf{\Sigma_1}\right), \mathcal{N}\left(\mathbf{z}^2 + \delta_2, \mathbf{\Sigma_2}\right) \right)$, according to lemma A.3, we can obtain:

$$
\begin{aligned}
&P(O \in B) \\
&= P\left( \delta_1^T \mathbf{\Sigma_1}^{-1}(\mathcal{N}\left(\mathbf{z}^1 + \delta_1, \mathbf{\Sigma_1}\right) - \mathbf{z}^1) + \delta_2^T \mathbf{\Sigma_2}^{-1}(\mathcal{N}\left(\mathbf{z}^2 + \delta_2, \mathbf{\Sigma_2}\right) - \mathbf{z}^2) \geq \|\delta_1^T \mathbf{\Sigma_1}^{-1} \mathbf{B_1} + \delta_2^T \mathbf{\Sigma_2}^{-1} \mathbf{B_2}\| \Phi^{-1}\left(\underline{p}\right) \right) \\
&= P\left( \delta_1^T \mathbf{\Sigma_1}^{-1} \mathcal{N}\left(\delta_1, \mathbf{\Sigma_1}\right) + \delta_2^T \mathbf{\Sigma_2}^{-1} \mathcal{N}\left(\delta_2, \mathbf{\Sigma_2}\right) \geq \|\delta_1^T \mathbf{\Sigma_1}^{-1} \mathbf{B_1} + \delta_2^T \mathbf{\Sigma_2}^{-1} \mathbf{B_2}\| \Phi^{-1}\left(\underline{p}\right) \right) \\
&= P\left( \delta_1^T \mathbf{\Sigma_1}^{-1}(\mathbf{B_1} \mathcal{N}\left(0, I\right) + \delta_1) + \delta_2^T \mathbf{\Sigma_2}^{-1}(\mathbf{B_2} \mathcal{N}\left(0, I\right) + \delta_2) \geq \|\delta_1^T \mathbf{\Sigma_1}^{-1} \mathbf{B_1} + \delta_2^T \mathbf{\Sigma_2}^{-1} \mathbf{B_2}\| \Phi^{-1}\left(\underline{p}\right) \right) \\
&= P\left( \|\delta_1^T \mathbf{\Sigma_1}^{-1} \mathbf{B_1} + \delta_2^T \mathbf{\Sigma_2}^{-1} \mathbf{B_2}\| \mathcal{N}\left(0, 1\right) + \delta_1^T \mathbf{\Sigma_1}^{-1} \delta_1 + \delta_2^T \mathbf{\Sigma_2}^{-1} \delta_2 \geq \|\delta_1^T \mathbf{\Sigma_1}^{-1} \mathbf{B_1} + \delta_2^T \mathbf{\Sigma_2}^{-1} \mathbf{B_2}\| \Phi^{-1}\left(\underline{p}\right) \right) \\
&= P\left( \mathcal{N}\left(0, 1\right) \geq \Phi^{-1}\left(\underline{p}\right) - \frac{\delta_1^T \mathbf{\Sigma_1}^{-1} \delta_1 + \delta_2^T \mathbf{\Sigma_2}^{-1} \delta_2}{\|\delta_1^T \mathbf{\Sigma_1}^{-1} \mathbf{B_1} + \delta_2^T \mathbf{\Sigma_2}^{-1} \mathbf{B_2}\|} \right) \\
&= \Phi\left( -\Phi^{-1}\left(\underline{p}\right) + \frac{\delta_1^T \mathbf{\Sigma_1}^{-1} \delta_1 + \delta_2^T \mathbf{\Sigma_2}^{-1} \delta_2}{\|\delta_1^T \mathbf{\Sigma_1}^{-1} \mathbf{B_1} + \delta_2^T \mathbf{\Sigma_2}^{-1} \mathbf{B_2}\|} \right).
\end{aligned}
$$

$\square$

---

**Algorithm 1** Certified robustness for visual GM.

---

**Input**: $(\mathbf{c}^1, \mathbf{c}^2, \mathbf{z}^1, \mathbf{z}^2)$; base function $f$; original $\sigma$; sample times $k_0$; similarity threshold $s$; number of copies $n$; regularization hyperparameter $\beta$.
**Output**: Core output $\mathbf{X}_c$; evaluation results.

1: Use the data augmentation and regularization term in Sec. 4.3 to train a visual GM model, and then obtain function $f_0$.
2: Obtain $\mathbf{B}_1, \mathbf{B}_2, \boldsymbol{\Sigma}_1, \boldsymbol{\Sigma}_2$ described in Sec. 4.2 for perturbing keypoint position pair or image pair using an optimization algorithm, and then obtain function $g_0$.
3: Sample $k_0$ number of input samples. For example, when perturbing keypoint position pair, we obtain the series: $\{(\mathbf{z}_1^{1'}, \mathbf{z}_1^{2'}), \ldots, (\mathbf{z}_{k_0}^{1'}, \mathbf{z}_{k_0}^{2'})\}$, where $\mathbf{z}_i^{1'} \sim \mathcal{N}(\mathbf{z}^1, \boldsymbol{\Sigma}_1)$ and $\mathbf{z}_i^{2'} \sim \mathcal{N}(\mathbf{z}^2, \boldsymbol{\Sigma}_2)$.
4: Predict the core output $\mathbf{X}_c$ and obtain the subspace $\mathcal{X}'$ using the first sampling series.
5: Sample $k = 10k_0$ number of input samples. For example, when perturbing the keypoint position pair, we obtain the series: $\{(\mathbf{z}_1^1, \mathbf{z}_1^2), \ldots, (\mathbf{z}_k^1, \mathbf{z}_k^2)\}$, where $\mathbf{z}_i^1 \sim \mathcal{N}(\mathbf{z}^1, \boldsymbol{\Sigma}_1)$ and $\mathbf{z}_i^2 \sim \mathcal{N}(\mathbf{z}^2, \boldsymbol{\Sigma}_2)$.
6: Calculate one-sided confidence lower bound $\underline{p}$ in Eq. 8 using the second sampling series.
7: **if** $\underline{p} < \frac{1}{2}$ **then**
8:   This input cannot be robustly certified.
9: **else**
10:   Obtain the sampling evaluation result and marginal radii evaluation result as in Sec. 4.4.
11: **end if**
12: **return** $\mathbf{X}_c$, evaluation results.

---

Table 2: Summary of main existing literature in learning GM.

| Method | Introduction |
|---|---|
| GMN (Zanfir & Sminchisescu, 2018) | The seminal work that employs a convolutional neural network to extract node features and constructs an end-to-end model with spectral matching. |
| PCA-GM (Wang et al., 2019) | Leveraging intra-graph and cross-graph structural information using graph convolutional networks. |
| CIE-H (Yu et al., 2019a) | Enhancing end-to-end training by edge embedding and Hungarian-based attention mechanism. |
| NGMv2 (Wang et al., 2021) | Developing a matching-aware graph convolution scheme with Sinkhorn iteration. |
| ASAR (Ren et al., 2022) | An appearance-aware regularizer is employed to explicitly increase the dissimilarities between similar keypoints and improve model robustness through adversarial attacks. |
| COMMON (Lin et al., 2023) | Integrating the momentum distillation strategy to balance the quadratic contrastive loss and reduce the impact of bi-level noisy correspondence. |

## B  SUMMARY OF RELATED METHODS

To present various methods of graph matching and certified robustness more clearly, we have categorized and reviewed the mainstream methods. Deep visual GM solvers aim to align the corresponding keypoints from different images based on node-to-node and edge-to-edge correlations. We introduce and compare the mainstream methods in Tab. 2. We then present some of the representative RS-based methods for certified robustness in Tab. 3, along with their applicable scenarios and features.

## C  METHODOLOGY SUPPLEMENT

In this section, we provide a supplement to the method described in Sec. 4. We first present the algorithm of the entire process, and then explain the construction and optimization of the joint Gaussian distribution under pixel perturbations.

Table 3: Summary of main existing literature in RS-type methods for robustness certification.

| Method | Introduction |
|---|---|
| **RS (Cohen et al., 2019)** | A pioneering work on certified robustness for classification tasks, demonstrating that Gaussian smoothing distributions can provide a provable $\ell_2$ perturbation bound. |
| **DSSN (Levine & Feizi, 2021)** | Providing a novel non-additive smoothing robustness certificate for the $\ell_1$ threat model. |
| **Median Smoothing (Chiang et al., 2020)** | Developing a new variant of smoothing specifically for detection based on the medians of the smoothed predictions. |
| **RS for Segmentation (Fischer et al., 2021)** | Presenting a scalable certification method for image and point cloud segmentation based on randomized smoothing. |
| **RS for Community Detection (Jia et al., 2020)** | Building a new smoothed community detection method via randomly perturbing the graph structure. |

---

**Algorithm 2** Algorithm for optimization.

---

**Input**: $L$ data; base function $f$; original $\sigma$; original $b$; iteration times $K$.
**Output**: $\mathbf{B}_1, \mathbf{B}_2, \mathbf{\Sigma}_1, \mathbf{\Sigma}_2$.
1: Initialize: $\sigma^0 \leftarrow \sigma, b^0 \leftarrow b$.
2: **for** $k = 0 \ldots K - 1$ **do**
3:     Calculate $\mathbf{B}_1^k, \mathbf{B}_2^k, \mathbf{\Sigma}_1^k, \mathbf{\Sigma}_2^k$ using $\sigma^k$ and $b^k$ according to Sec. 4.
4:     Initialize the sum of optimization goal $O$.
5:     **for** $l = 0 \ldots L - 1$ **do**
6:         Initialize $k^{th}$ data.
7:         Sample $\varepsilon_1 \sim \mathcal{N}(0, \mathbf{B}_1^k), \varepsilon_2 \sim \mathcal{N}(0, \mathbf{B}_2^k)$.
8:         Calculate $p$ according to Eq. 8 and eigenvalues of $\mathbf{B}_1^k, \mathbf{B}_2^k$, then calculate the optimization goal $O^l$ as in Eq. 10.
9:         $O \leftarrow O + O^l$.
10:     **end for**
11:     $\sigma^{k+1}, b^{k+1} \leftarrow \nabla_{\sigma^k, b^k} O$.
12: **end for**
13: Calculate $\mathbf{B}_1, \mathbf{B}_2, \mathbf{\Sigma}_1, \mathbf{\Sigma}_2$ using $\sigma^{K-1}$ and $b^{K-1}$ according to Sec. 4.
14: **return** $\mathbf{B}_1, \mathbf{B}_2, \mathbf{\Sigma}_1, \mathbf{\Sigma}_2$.

---

### C.1 ALGORITHM OF THE ENTIRE PROCESS

Alg. 1 consists of training and testing parts. In the training part, we use data augmentation and a regularization term based on the output similarity as Sec. 4.3 to train a model. In the testing part, we employ Monte Carlo sampling to estimate the certification result in practice. First, we construct and optimize the smoothing joint Gaussian distribution according to Sec. 4.2 and construct the smoothed model $g_0$. Second, we sample $(\varepsilon_1, \varepsilon_2)$ with $k_0$ times and obtain the core output $\mathbf{X}_c$ in Eq. 4 and subspace $\mathcal{X}'$ in Eq. 5. Then we sample $(\varepsilon_1, \varepsilon_2)$ with $k$ times, and count how many outputs fall into the subspace $\mathcal{X}'$ to obtain the probability $p$ in Eq. 8 and the certified space in Eq. 9. Finally, we use two quantitative methods as in Sec. 4.4 to obtain evaluation results.

Alg. 2 summarizes the updates for optimizing by solving Eq. 10 with $K$ steps of stochastic gradient ascent. $p$ is estimated by the Monte Carlo sampling algorithm in the subsequent certification process, but we simplify its estimation in the optimization algorithm. Since we do not need a very precise $p$ value here, but a favorable trend, we calculate it by sampling only once. This approach not only improves the efficiency of the optimization algorithm but also avoids the high variance in the gradient estimation caused by multiple sampling. We fix the number of iterations for all optimization algorithms to $K = 10$, the size of data used for optimization to $L = 100$, and set the original correlation parameter $b = 0.01$. Therefore, the entire optimization process is relatively fast and can be relatively easily applied to various visual GM models.

## C.2 Smoothing Distribution for Perturbing Image Pixels

Due to the large number of image pixels, which far exceeds the number of keypoints, constructing correlation matrices between pixel points as in Sec. 4.2 is computationally expensive, not to mention that mining the correlation between pixels is not trivial. We, therefore, simplify $\mathbf{B}$ to $\sigma$ as in Cohen et al. (2019) under pixel perturbations and modify the optimization problem accordingly:

$$\arg\max_{\sigma} \Phi^{-1}\left(\underline{p}\right)\sigma. \tag{25}$$

# D Additional experiment settings

This section provides the details of the baseline for certification, GM solvers, which are supplementary to Sec. 5.1.

## D.1 Baseline for Certification

This paper adopts modified RS (Cohen et al., 2019) as the baseline method for the proposed CR-OSRS strategy, which is referred as RS-GM. Unless otherwise stated, we follow the same experimental parameter settings as RS. We use the hypothesis test (Hung & Fithian, 2019) as in Cohen et al. (2019) by using $\alpha$ to represent the probability of obtaining incorrect matching results. In this study, we set $\alpha = 0.001$, which ensures a high probability (99.9%) of certification. $\alpha$ can be arbitrarily small, so in theory our method is highly reliable. We choose the Monte Carlo sample number $k$ in Alg. 1 to be 1000, which is smaller than the sample number for classifier certification, due to the low efficiency of the GM solver. Theoretically, increasing $k$ would improve the certification results, but at the expense of the efficiency of the GM solver. We reveal the impact of different $k$ on the experimental results in Appendix E.1.

## D.2 Deep Graph Matching Solvers

This paper evaluates the proposed method on the Pascal VOC dataset (Everingham et al., 2010) with Berkeley annotations (Bourdev & Malik, 2009), the Willow ObjectClass dataset (Cho et al., 2013) and SPair-71k dataset (Min et al., 2019) for visual graph matching. Following the protocol of Wang et al. (2021), for the Pascal VOC dataset, we exclude images with poor annotations. Then we use 100 inputs (about 650 keypoints) from 20 categories in the dataset to test the proposed method on six representative deep GM methods: GMN (Zanfir & Sminchisescu, 2018), PCA-GM (Wang et al., 2019), CIE-H (Yu et al., 2019a), NGMv2 (Wang et al., 2021), ASAR (Ren et al., 2022) and COMMON (Lin et al., 2023), using the checkpoints of these GM models provided by ThinkMatch (`https://github.com/Thinklab-SJTU/ThinkMatch`). For the Willow ObjectClass dataset, we use 100 inputs from 5 categories to test the method on the NGMv2 solver. For the SPair-71k dataset, we use 90 inputs from 5 categories to test the method on the NGMv2 solver.

# E Experimental Results

This section first presents the certification results on the Willow ObjectClass dataset and SPair-71k dataset, then presents the certification results for ASAR (Ren et al., 2022) and COMMON (Lin et al., 2023) solvers. Finally, we report additional results on how the parameters $n$, $\kappa$, $\beta$, and $k$ affect the certified robustness and model performance in Appendix E.1. Finally, it shows the results under pixel perturbations in Appendix E.2.

## E.1 Additional Experimental Results under Keypoint Position Perturbations

First, we investigate the relationship of CA and three marginal radii ($\|\delta\|_{\text{lower}}$, $\|\delta\|_{\text{upper}}$, and $\|\delta\|_{\text{volume}}$) for RS-GM and CR-OSRS on the Willow ObjectClass dataset in Fig. 4 and the SPair-71k dataset Fig. 5. We also compare the results of adding only data augmentation or adding both data augmentation and the regularization term, as shown in Eq. 11. In Fig. 4, the curve of CR-OSRS is almost always above RS-GM, indicating that CR-OSRS corresponds to larger radii for the same certified accuracy

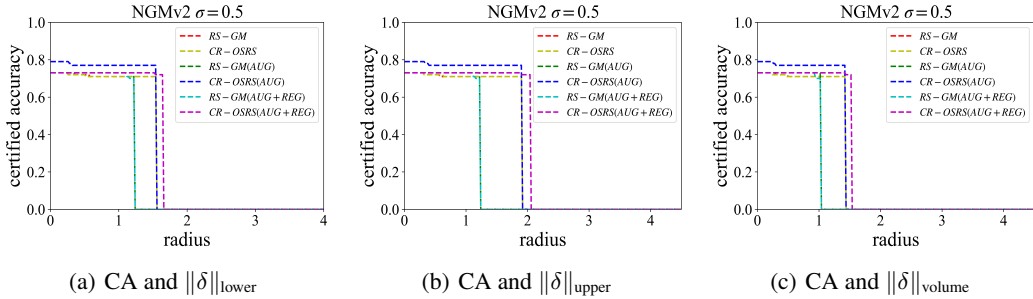

(a) CA and $\|\delta\|_{\text{lower}}$      (b) CA and $\|\delta\|_{\text{upper}}$      (c) CA and $\|\delta\|_{\text{volume}}$

Figure 4: Certified accuracy (CA) achieved by RS-GM and CR-OSRS for NGMv2 on Willow ObjectClass dataset when perturbing keypoint positions. Fig. 4 shows the result with original $\sigma = 0.5$, $s = 0.9$ in Eq. 5, $\beta = 0.01$ and $n = 2$ in Eq. 11.

and corresponds to higher accuracy for the same radii, which implies greater certified robustness. However, we observe that the improvement of model performance by data augmentation and the regularization term is not as significant as on the Pascal VOC dataset. We conjecture that this is because Willow is less sensitive to perturbations for keypoint positions. Therefore, data augmentation and the regularization term have little effect on the "majority decision" of RS and even cause the model to underfit. In Fig. 5, The curve of CR-OSRS is almost always above RS-GM, which implies greater certified robustness and matching accuracy. At the same time, it also illustrates that the proposed data augmentation and the regularization term are effective. As Fig. 4, Fig. 5 and Table 1 show that our method can be applied to various datasets and GM solvers.

Second, we examine the relationship of CA and three marginal radii for RS-GM and CR-OSRS on ASAR (Ren et al., 2022) and COMMON (Lin et al., 2023) in Fig. 6 and Fig. 7. We also compare the results of adding only data augmentation, as well as adding both the data augmentation and the regularization term, as in Eq. 11. The curve of CR-OSRS is almost always above RS-GM, which implies greater certified robustness and matching accuracy. At the same time, it also demonstrates that the proposed data augmentation and the regularization term are effective.

Third, we further examine the effect of the number of copies $n$, the constraint hyperparameter $\kappa$, the regularization hyperparameter $\beta$ in Eq. 11 as well as the Monte Carlo sample number $k$ for Monte Carlo sampling on the certification results, which were not examined in Sec. 5.2. We vary $n$ from $n \in \{1, 2, 3, 4\}$ and plot the certified accuracy with each $n$ in Fig. 8. Choosing appropriate values of $n$ is crucial for improving the model performance. We vary $\kappa$ from $\kappa \in \{0, \frac{1}{300}, \frac{1}{200}, \frac{1}{100}\}$ and plot the certified accuracy with each $\kappa$ in Fig. 9. The figure shows that $\kappa$ had little overall influence on the outcomes, but a larger $\kappa$ results in a larger $\|\delta\|_{\text{volume}}$ and $\|\delta\|_{\text{upper}}$ as well as a smaller $\|\delta\|_{\text{lower}}$. We vary $\beta$ from $\beta \in \{0.005, 0.01, 0.02\}$ and plot the certified accuracy with each $\beta$ in Fig. 10. Choosing appropriate values of $\beta$ will help balance the trade-off between matching performance and certified robustness. Furthermore, we vary $k$ from $k \in \{1000, 2000, 3000, 4000, 5000\}$ and plot the certified accuracy with each $k$ in Fig. 11, which projects how the certified accuracy would change when using more samples $k$ (under the assumption $k = 10k_0$). We observe that when $k$ increases, the robustness can be certified to be stronger, which is influenced by the Monte Carlo sampling algorithm.

### E.2 EXPERIMENTAL RESULTS ON IMAGE PIXEL PERTURBATIONS

For perturbing image pixels, we plot the relationship of certified accuracy (CA) and three marginal radii ($\|\delta\|_{\text{lower}}$, $\|\delta\|_{\text{upper}}$, and $\|\delta\|_{\text{volume}}$) in Fig. 12 with the original $\sigma = 0.5$, $\beta = 0.01$ and $n = 2$. As discussed in Section C.2, constructing a correlation matrix between pixels is computationally expensive due to the large number of image pixels. Moreover, it is challenging to extract the correlation between pixels. Hence, we employ RS-GM to achieve robustness certification under pixel perturbations. Fig. 12 demonstrates the effectiveness of data augmentation and the regularization term. Data augmentation has a significant effect, but the regularization term does not improve performance in this case. We conjecture that this is because the output distribution of a fixed datum sample is not too dispersed under multiple perturbations, so the regularization term has negligible impact.

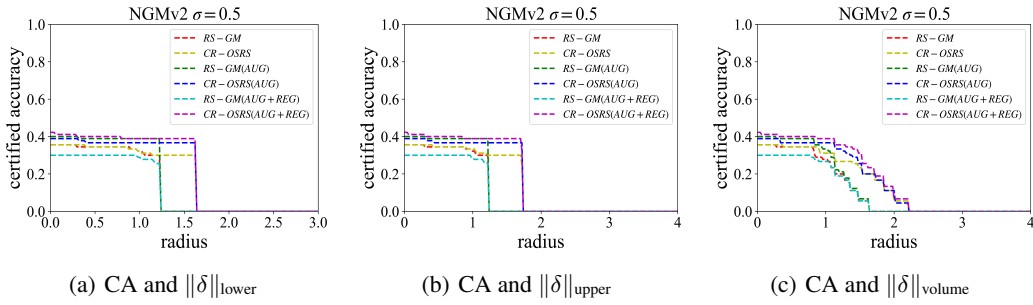

(a) CA and $\|\delta\|_{\text{lower}}$     (b) CA and $\|\delta\|_{\text{upper}}$     (c) CA and $\|\delta\|_{\text{volume}}$

Figure 5: Certified accuracy (CA) achieved by RS-GM and CR-OSRS for NGMv2 on SPair-71k dataset when perturbing keypoint positions. Fig. 5 shows the result with original $\sigma = 0.5$, $s = 0.9$ in Eq. 5, $\beta = 0.01$ and $n = 2$ in Eq. 11.

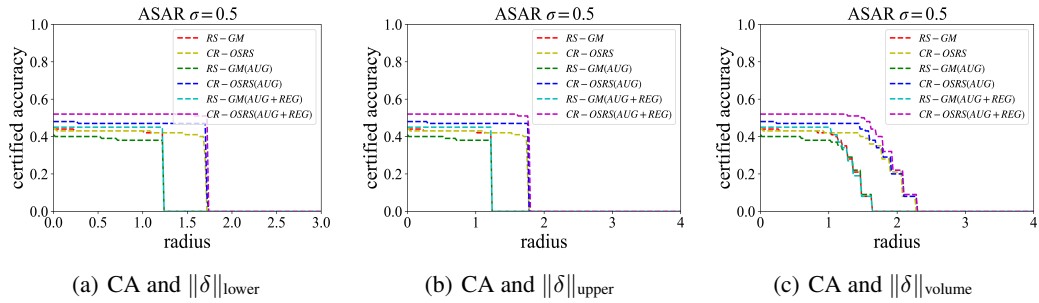

(a) CA and $\|\delta\|_{\text{lower}}$     (b) CA and $\|\delta\|_{\text{upper}}$     (c) CA and $\|\delta\|_{\text{volume}}$

Figure 6: CA achieved by CR-OSRS and RS-GM for ASAR on Pascal VOC when perturbing keypoint positions. "AUG" denotes data augmentation and "REG" denotes the regularization term in Eq. 11. Fig. 6 shows the result for original $\sigma = 0.5$, $s = 0.9$ in Eq. 5, $\beta = 0.01$ and $n = 2$ in Eq. 11.

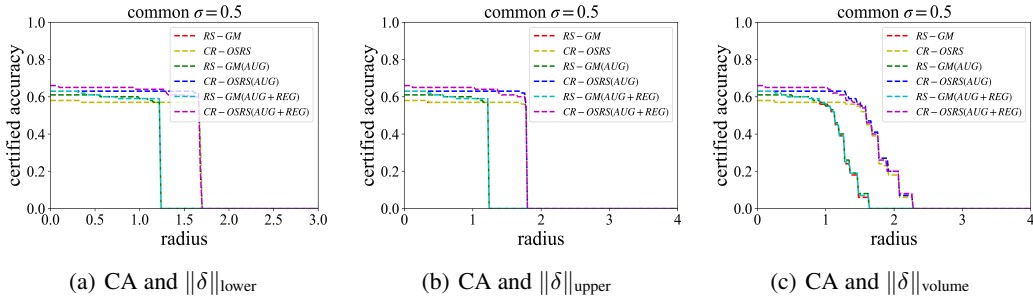

(a) CA and $\|\delta\|_{\text{lower}}$     (b) CA and $\|\delta\|_{\text{upper}}$     (c) CA and $\|\delta\|_{\text{volume}}$

Figure 7: CA achieved by CR-OSRS and RS-GM for COMMON on Pascal VOC when perturbing keypoint positions. "AUG" denotes data augmentation and "REG" denotes the regularization term in Eq. 11. Fig. 7 shows the result for original $\sigma = 0.5$, $s = 0.9$ in Eq. 5, $\beta = 0.01$ and $n = 2$ in Eq. 11.

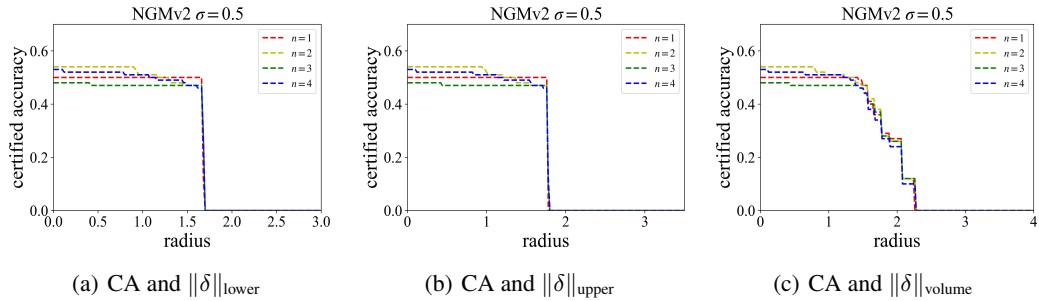

(a) CA and $\|\delta\|_{\text{lower}}$      (b) CA and $\|\delta\|_{\text{upper}}$      (c) CA and $\|\delta\|_{\text{volume}}$

Figure 8: Projections for the certified accuracy if the loss function parameter $n$ had been larger or smaller. Fig. 8 shows the result for CR-OSRS trained by data augmentation and regularization term with $\sigma = 0.5$, $s = 0.9$ and $\beta = 0.01$ for NGMv2 on Pascal VOC.

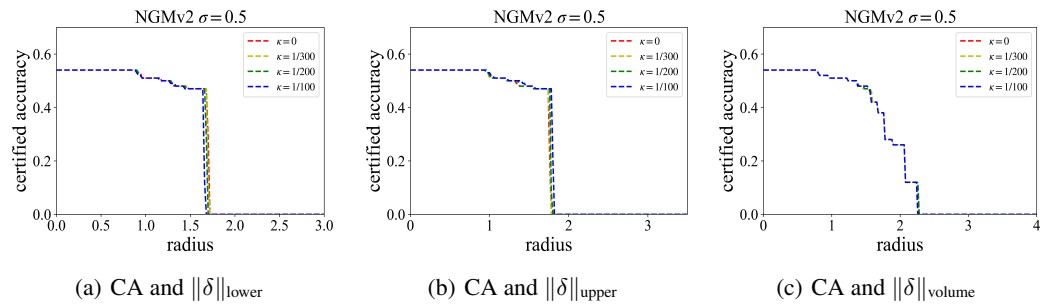

(a) CA and $\|\delta\|_{\text{lower}}$      (b) CA and $\|\delta\|_{\text{upper}}$      (c) CA and $\|\delta\|_{\text{volume}}$

Figure 9: Projections for the certified accuracy if the constraint hyperparameter $\kappa$ had been larger or smaller. Fig. 9 shows the result for CR-OSRS trained by data augmentation and regularization term with $\sigma = 0.5$, $s = 0.9$ and $\beta = 0.01$ for NGMv2 on Pascal VOC.

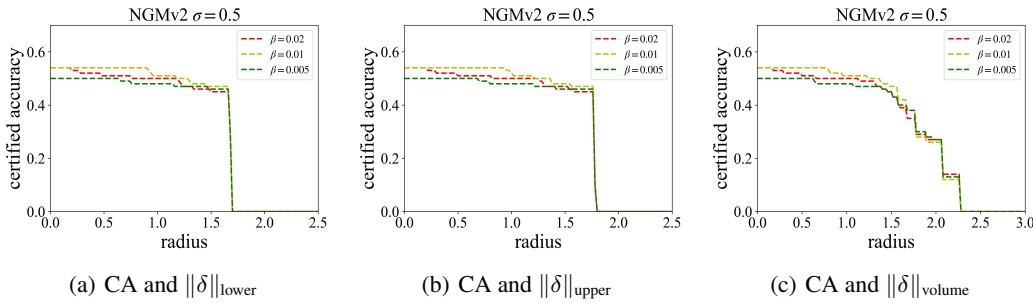

(a) CA and $\|\delta\|_{\text{lower}}$      (b) CA and $\|\delta\|_{\text{upper}}$      (c) CA and $\|\delta\|_{\text{volume}}$

Figure 10: Projections for the certified accuracy if the regularization hyperparameter $\beta$ had been larger or smaller. Fig. 10 shows the result for CR-OSRS trained by data augmentation and regularization term with $\sigma = 0.5$, $s = 0.9$ and $n = 2$ for NGMv2 on Pascal VOC.

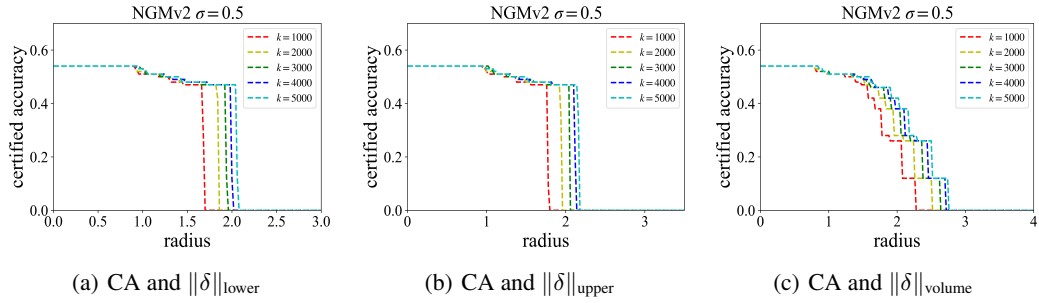

(a) CA and $\|\delta\|_{\text{lower}}$    (b) CA and $\|\delta\|_{\text{upper}}$    (c) CA and $\|\delta\|_{\text{volume}}$

Figure 11: Projections for the certified accuracy if the Monte Carlo sample number $k$ had been larger or smaller. Fig. 11 shows the result for CR-OSRS trained by data augmentation and regularization term with $\sigma = 0.5$, $s = 0.9$, $\beta = 0.01$ and $n = 2$ for NGMv2 on Pascal VOC.

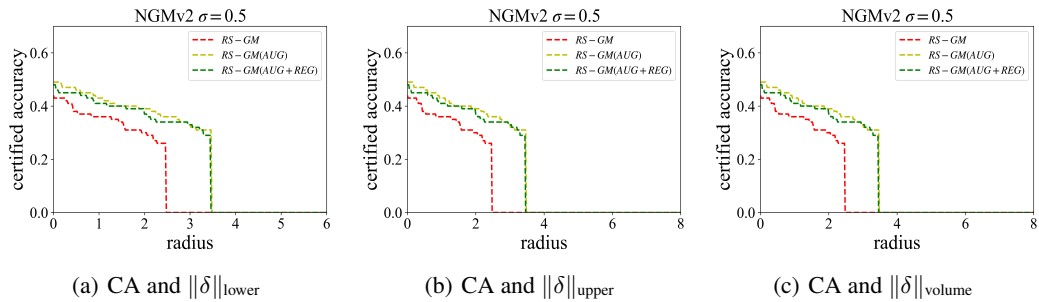

(a) CA and $\|\delta\|_{\text{lower}}$    (b) CA and $\|\delta\|_{\text{upper}}$    (c) CA and $\|\delta\|_{\text{volume}}$

Figure 12: Certified accuracy (CA) achieved by RS-GM for NGMv2 on Pascal VOC under pixel perturbations.

