# OpenReview forum: "Certified Robustness on Visual Graph Matching via Searching Optimal Smoothing Range"
_ICLR.cc/2024/Conference — Submitted to ICLR 2024_

### Official Review · Reviewer_4FJt · 2023-10-26

**Soundness:** 3 good
**Presentation:** 3 good
**Contribution:** 3 good
**Rating:** 8
**Confidence:** 4

**Summary:**

This paper proposes a certified robustness method of visual graph matching (GM) against adversarial perturbations on image pixels and keypoint positions. The method, named CR-OSRS, uses a joint Gaussian distribution to construct a smoothed model and searches for the optimal smoothing range that balances the trade-off between certified robustness and matching performance. The paper also introduces a data augmentation technique and a regularization term to improve the model performance during training. The paper provides theoretical analysis and empirical evaluation of the proposed method on two GM datasets and four GM solvers.

**Strengths:**

- The paper proposes a principled method that leverages the correlation between keypoints to construct a joint smoothing distribution and uses global optimization to find the optimal smoothing range.
- The paper provides rigorous theoretical analysis and proofs for the certified robustness guarantee, as well as two methods to quantify the certified space.
- The paper conducts extensive experiments on two GM datasets and four GM solvers, and demonstrates the effectiveness and superiority of the proposed method over the baseline method.

**Weaknesses:**

- The paper lacks sufficient details regarding the implementation of the optimization algorithm for determining the optimal smoothing range, specifically the step 2 in Algorithm 1. Clarity is needed on the efficiency and scalability of this algorithm, especially in the context of larger-scale problems.
- The literature review on graph matching and its robustness omits references to recent works on noisy correspondence in graph matching [2], which is closely related to the issue of adversarial attacks.
- It is advisable to conduct a comparison between the proposed method and other existing techniques for robust GM, such as ASAR[1] and COMMON [2]. Specifically, COMMON addresses robust graph matching by considering noisy correspondence during training, while ASAR takes adversarial attacks into account during training. Evaluating these methods alongside CR-OSRS would provide more comprehensive experimental insights. Furthermore, reporting the certified accuracy and average certified radius for these models is encouraged.
- Since the author outlines four challenges in the Introduction, it would be beneficial to emphasize these points within the Method section, using C1 to C4.

**Questions:**

- Could you provide more details on the global optimization algorithm used to determine the optimal smoothing range? What is the computational complexity, and how scalable is this algorithm?
- The choice of baseline methods and datasets in your evaluation appears somewhat dated (prior to 2021). Would it be possible to include the recent and popular Spair-71k [3] graph matching dataset in your experiments to provide more up-to-date and comprehensive results?

The primary focus of the rebuttal should be on addressing the concerns mentioned in the "Weaknesses" section, particularly by providing further experiments involving different methods and a larger dataset.

[1] Appearance and Structure Aware Robust Deep Visual Graph Matching: Attack, Defense and Beyond, CVPR 2022

[2] Graph Matching with Bi-level Noisy Correspondence, ICCV 2023

[3] SPair-71k: A Large-scale Benchmark for Semantic Correspondence, arxiv 2019

---

> ### Author Response · Authors · 2023-11-19
> **Response to Reviewer 4FJt (1/2)**
>
> >***Q1: The paper lacks sufficient details regarding the implementation of the optimization algorithm for determining the optimal smoothing range, specifically the step 2 in Algorithm 1. Clarity is needed on the efficiency and scalability of this algorithm, especially in the context of larger-scale problems.
> Could you provide more details on the global optimization algorithm used to determine the optimal smoothing range? What is the computational complexity, and how scalable is this algorithm?***
> >
> **R1:** Thank you for your question. We provide more details of the optimization algorithm below. We have also revised the paper accordingly, please refer to Appendix.C in the updated version for more details.
>
> * ***We first elaborate on the motivation for designing the optimization algorithm.*** As described in Sec.4.2, we employ a smoothing distribution $\mathbf{B}$ with correlation to transform the basic GM model into a one with certified robustness. However, simply setting the values of $\mathbf{B}_1$ and $\mathbf{B}_2$ in advance may not achieve satisfactory results. Therefore, we propose to optimize $\mathbf{B}_1$ and $\mathbf{B}_2$ around the initial values to enhance the model’s certified robustness. We optimize $\sigma$ and $b$ for the dataset, irrespective of the network parameters which are kept constant here.
> * ***Second, we describe the design rationale and implementation of the optimization algorithm.*** As mentioned in Sec.4.4, the proxy radius is a measure of the certified space, so our optimization algorithm aims to maximize the proxy radius on the dataset. Moreover, we enforce a constraint on $b$ in Eq.10, which is motivated by two main reasons. One is that $b$ has to be positive, since a negative b would invalidate the smoothing distribution; the other is to balance the trade-off between certified robustness and model performance. A natural solver for Eq.10 is stochastic gradient ascent with the expectation approximated by Monte Carlo samples. Thus, the gradient of the objective at the $k^{th}$ iteration can be estimated as follows:
> $$
> \nabla_{\sigma^k, b^k} \
> \Phi^{-1}\left(\underline{p}\right)\sum_{i=1,2}\left(\sqrt[2m_{i}]{\prod_{j}^{m_i} \lambda_{ij}}\right)+\kappa b^k.
> $$
> In this work, $\underline{p}$ is estimated by the Monte Carlo sampling algorithm in the subsequent certification process, but we simplify its estimation in the optimization algorithm. Since we do not need a very precise $\underline{p}$ value here, but a favorable trend, we calculate it by sampling only once. This approach not only improves the efficiency of the optimization algorithm but also avoids the high variance in the gradient estimation caused by multiple sampling.
> * In this work, we fix the number of iterations for all optimization algorithms to $K=10$, and the size of data used for optimization to 100. Therefore, the entire optimization process is relatively fast, with each optimization process taking less than 2 minutes on CPU (Intel(R) Core(TM) i7-7820X CPU @ 3.60GHz) and GPU (GTX 2080 Ti GPU). ***Consequently, the algorithm can be relatively easily applied to various visual GM models.*** We summarize the updates for optimizing by solving Eq.10 with $K$ steps of stochastic gradient ascent in Algorithm.2 in the paper, please refer to the revised version.

---

> ### Author Response · Authors · 2023-11-19
> **Response to Reviewer 4FJt (2/2)**
>
> >***Q2: It is advisable to conduct a comparison between the proposed method and other existing techniques for robust GM, such as ASAR[1] and COMMON [2]. Specifically, COMMON addresses robust graph matching by considering noisy correspondence during training, while ASAR takes adversarial attacks into account during training. Evaluating these methods alongside CR-OSRS would provide more comprehensive experimental insights. Furthermore, reporting the certified accuracy and average certified radius for these models is encouraged.***
> >
> **R2:**
> We appreciate your valuable feedback. We have performed additional experiments based on your suggestions and further elaborated our proposed certified robustness method.
> * The ASAR and COMMON methods you referred to are the recently proposed visual GM methods with relatively good results. We calculated CR (in link: https://anonymous.4open.science/r/Certified-Robustness-on-Visual-Graph-Matching-via-Searching-Optimal-Smoothing-Range-E5F0/Rebuttal/Figure/ASAR/ and https://anonymous.4open.science/r/Certified-Robustness-on-Visual-Graph-Matching-via-Searching-Optimal-Smoothing-Range-E5F0/Rebuttal/Figure/COMMON) and ACR (in table below) achieved by CR-OSRS and RS-GM on Pascal VOC under keypoint position perturbations for these two models. ***The results indicate that the certified robustness obtained by CR-OSRS is also superior on these two basic models.*** This result has been updated in the new version of the paper, please refer to Tab.1, Fig.6, Fig.7 and Appendix.E.1 in the paper.
>
> Table: ACR achieved by CR-OSRS and RS-GM for six GM solvers on Pascal VOC under keypoint position perturbations.
>
> |     | CR-OSRS+AUG+REG      | CR-OSRS+AUG+REG      | CR-OSRS+AUG+REG       | RS-GM+AUG+REG        | RS-GM+AUG+REG        | RS-GM+AUG+REG         |
> | --- | ----------------------- | ----------------------- | ------------------------ | ----------------------- | ----------------------- | ------------------------ |
> |     | $\|\|\delta_{lower}\|\|$ | $\|\|\delta_{upper}\|\|$ | $\|\|\delta_{volume}\|\|$ | $\|\|\delta_{lower}\|\|$ | $\|\|\delta_{upper}\|\|$ | $\|\|\delta_{volume}\|\|$ |     |
> COMMON| 1.550|1.751|1.900|0.952|0.952|1.069|
> ASAR|1.541|1.648|1.968|0.683|0.683|0.841|
> NGMv2| 1.425|1.586|1.934|0.778| 0.778|1.010|
> CIE-H|0.987|1.167|1.354|0.572 |0.572 | 0.731|
> PCA-GM|0.954|1.158|1.340|0.546|0.546|0.686|
> GMN|0.899|1.076|1.253|0.514| 0.514|0.617|
>
> * ***We offer more explanation on the difference between certified robustness and empirical robustness as well as the significance of CR-OSRS.*** The difference is that the results obtained by certified robustness are rigorously verifiable, that is, when the perturbation is within a certified range, the output is always invariant. However, empirical robustness does not have this theoretical guarantee, which means that it may have well defense capabilities against current attack methods, but it is highly susceptible to future unseen attacks. Therefore, the two types of robustness cannot be directly compared. This paper uses CR-OSRS to transform basic solvers into smoothed models with certified robustness for comparison and analysis. We also observe, in practice, that the better the matching performance of the basic model, the better the certified robustness and matching performance of the smoothed model will be. For instance, COMMON method enhances the model performance by leveraging noisy correspondence, which will make the smoothed models obtained by CR-OSRS more certifiably robust.
>
> >***Q3: Since the author outlines four challenges in the Introduction, it would be beneficial to emphasize these points within the Method section, using C1 to C4.***
> >
> **R3:** Thanks for your suggestion, which will improve the clarity of our paper. We have implemented the corresponding changes in the new version of the paper.
>
> >***Q4: The choice of baseline methods and datasets in your evaluation appears somewhat dated (prior to 2021). Would it be possible to include the recent and popular Spair-71k [3] graph matching dataset in your experiments to provide more up-to-date and comprehensive results?***
> >
> **R4:** We appreciate your suggestion, which will enhance the quality of our paper. We show the results on the dataset Spair-71k (in link: https://anonymous.4open.science/r/Certified-Robustness-on-Visual-Graph-Matching-via-Searching-Optimal-Smoothing-Range-E5F0/Rebuttal/Figure/SPair-71k). The figures show the CA achieved by CR-OSRS and RS-GM for NGMv2 on Spair-71k when perturbing keypoint positions. The figure displays the result for original $\sigma=0.5$, $s=0.9$ , $\beta=0.01$ and $n=2$. The curve of CR-OSRS is almost always above RS-GM, which implies greater certified robustness and matching accuracy. At the same time, it also illustrates that the proposed data augmentation and the regularization term are effective. This result has been updated in the new version of the paper, please refer to Fig.5 and Appendix.E.1 in the paper.

---

> ### Comment · Reviewer_4FJt · 2023-11-20
> **Good job**
>
> I appreciate the response from the authors. It would be beneficial if you could incorporate discussions on **certified robustness, empirical robustness (model robustness?), and adversarial robustness** (Point 2 of R2) into the paper. I'm happy to increase my rating.

---

> > ### Author Response · Authors · 2023-11-20
> > **A Thank-You Note to Reviewer 4FJt**
> >
> > Dear Reviewer 4FJt,
> >
> > We would like to express our sincere gratitude for your time and valuable suggestions. We are delighted that our response has addressed your concerns and earned your approval. We have incorporated the discussion of certified robustness and empirical robustness into the paper. Please refer to Sec.1 and Sec.5 in the updated version for more details.
> >
> > Thank you again for your effort and giving us the opportunity to improve our paper based on your insightful comments.
> >
> > Best regards.

---

### Official Review · Reviewer_wBHW · 2023-10-30

**Soundness:** 3 good
**Presentation:** 3 good
**Contribution:** 3 good
**Rating:** 6
**Confidence:** 3

**Summary:**

This work introduces a novel certified robustness algorithm for visual graph matching problem. To achieve a larger certified space as well as better trade-offs between certified robustness and matching performance, the authors propose to search the optimal joint Gaussian distribution for random smoothing with a subspace defined by a similarity threshold. The authors also provide theoretical analysis that the matching matrix can be bounded within the subspace. Extensive comparisons with various baselines demonstrate the effectiveness of the proposed algorithm.

**Strengths:**

1. The paper is well-organized and easy to follow.
2. The proposed algorithm is well motivated by the theoretical analysis.
3. The results are promising compared with other baselines.

**Weaknesses:**

My major concerns mainly lie in the ablation studies:

1. In Eq. 10, the authors mentioned that a constraint on b is imposed in the optimization. However, how this constraint works is not well explained. The effectiveness of this constraint is not evaluated in the experiments.
2. The authors introduced a regularization in Eq. 11, however, the ablation study of the variant without this regularization is missing.

**Questions:**

1. Please provide more details of the constraint in Eq. 10 as well as ablation studies.
2. Please provide the ablation study of the regularization in Eq. 11.

---

> ### Author Response · Authors · 2023-11-19
> **Response to Reviewer wBHW**
>
> >***Q1: In Eq. 10, the authors mentioned that a constraint on b is imposed in the optimization. However, how this constraint works is not well explained. The effectiveness of this constraint is not evaluated in the experiments. Please provide more details of the constraint in Eq. 10 as well as ablation studies.***
> >
> **R1:** Thank you for your questions and suggestions. We will clarify the role of the constraint and its ablation study. We have also revised the paper accordingly, please refer to Appendix.C and Appendix.E.1 in the updated version for more details.
> * ***We first explain the rationale behind the design of the optimization algorithm.*** As described in Sec.4.2, we employ a smoothing distribution $\mathbf{B}$ with correlation to transform the basic GM model into a one with certified robustness. However, simply setting the values of $\mathbf{B}_1$ and $\mathbf{B}_2$ in advance may not achieve satisfactory results. Therefore, we propose to optimize $\mathbf{B}_1$ and $\mathbf{B}_2$ around the initial values to enhance the model’s certified robustness. It is worthwhile to mention that we optimize $\sigma$ and $b$ on the dataset, regardless of the architecture and the training procedure of the basic model.
> * As mentioned in Sec.4, the proxy radius is a measure of the certified robustness of the algorithm, so our optimization algorithm aims to maximize the proxy radius on the dataset as shown in Eq.10. ***Moreover, we enforce a constraint on the correlation parameter $b$, which is motivated by two main reasons.*** One is that $b$ has to be positive, since a negative $b$ would invalidate the smoothing distribution; the other is to balance the trade-off between certified robustness and model performance. A large $b$ may improve the matching performance but reduce the certified space.
> * We have performed ablation experiments to examine the effect of parameter $\kappa$ on the certification results as in figures (https://anonymous.4open.science/r/Certified-Robustness-on-Visual-Graph-Matching-via-Searching-Optimal-Smoothing-Range-E5F0/Rebuttal/Figure/Ablation%20study%20of%20kappa/). The figures shows that $\kappa$ had little overall influence on the outcomes, but a larger $\kappa$ results in a larger $||\delta_{volume}||$ and $||\delta_{upper}||$ as well as a smaller $||\delta_{lower}||$. However, we still stress that the constraint plays a vital role in ensuring that the value of $b$ remains within a reasonable optimization range. This result has been updated in the new version of the paper, please refer to it in Appendix.E.1 and Fig.9.
>
> >***Q2: The authors introduced a regularization in Eq. 11, however, the ablation study of the variant without this regularization is missing. Please provide the ablation study of the regularization in Eq. 11.***
> >
> **R2:** Thank you for your question. We actually have presented the ablation study you referred to in Fig.2. We evaluate the certification results of RS-GM and CR-OSRS on the basic model, the model with data augmentation (AUG), and the model with data augmentation and the regularization term (AUG+REG). We also explain in Sec.5.2 that the regularization term is always related to data augmentation as shown in Eq.11. In the absence of data augmentation, the outputs corresponding to all copy data are identical and thus the regularization term is always zero. Fig.2 demonstrates that the proposed data augmentation and the regularization term are effective.

---

### Official Review · Reviewer_eczv · 2023-11-01

**Soundness:** 3 good
**Presentation:** 3 good
**Contribution:** 3 good
**Rating:** 6
**Confidence:** 4

**Summary:**

This paper addresses the challenge of robustness certification in visual graph matching, a problem lying between traditional regression models and combinatorial optimization. Two key technical innovations are introduced: using pairs of graph data as inputs and adding constraints to the output space. The authors propose an Optimal Smoothing Range Search approach, inspired by random smoothing techniques, to enhance robustness. Practical techniques such as output space partitioning based on similarity, data augmentation, and a similarity-based regularization term are also presented. Experimental results confirm the effectiveness of these methods.

**Strengths:**

1. The paper addresses an intriguing and essential problem, as existing certification methods are primarily geared toward image recognition, leaving structured prediction, especially combinatorial optimization, less explored. While graph matching is a well-studied problem in recent machine learning literature, certification in this context has been notably absent.

2. The novel techniques, particularly the global optimization search algorithm, stand out as a reasonable and innovative approach, well-suited to the new problem setting examined in this paper.

3. The paper introduces two new methods for measuring the certified space, offering valuable tools for quantitative analysis.

4. The paper achieves a commendable balance between matching accuracy and robustness certification, as evidenced by extensive experiments.

**Weaknesses:**

1. The presentation can be improved for better clarity, as it involves multiple areas ranging from graph matching (combinatorial optimization), robustness certification, visual recognition, etc.
2. the paper lacks some discussion for enlarging its potential impact to other combinatorial tasks or any limitation and difficulty to extend its adaption to other tasks.

**Questions:**

1. Have you explored the possibility of extending your approach to address problems beyond graph matching? Given the ubiquity of combinatorial optimization on graphs, a discussion on a potentially more general framework could be beneficial.

2. Can you add a table or figure to summarize and compare related methods from multiple aspects for better accessibility of readers?

---

> ### Author Response · Authors · 2023-11-19
> **Response to Reviewer eczv (1/3)**
>
> >***W1/Q2: The presentation can be improved for better clarity, as it involves multiple areas ranging from graph matching (combinatorial optimization), robustness certification, visual recognition, etc. Can you add a table or figure to summarize and compare related methods from multiple aspects for better accessibility of readers?***
> >
> **R1:** We appreciate your insightful suggestions. Our research draws on knowledge from multiple fields, and we aim to present and compare them more rigorously and clarify their relevance to this study according to your suggestions. We show the comparison of visual GM methods and certified robustness methods in Tab.1 and Tab.2 below. We have also revised the paper accordingly, please refer to Appendix.B in the updated version for more details.
>
> * **First**, we provide a brief introduction to visual Graph Matching(GM). GM is a fundamental problem in combinatorial optimization that involves identifying node correspondences between graphs. As introduced in Sec.3, deep visual GM deals with images with keypoints as inputs and solves in an end-to-end manner. Visual GM consists of three components: keypoint feature extractor, affinity learning, and final correspondence solver. ***We introduce and compare the main visual GM methods used in this work in Tab.1 below, which we have added to the paper, please refer to Appendix.B for more details.***
>
> Table 1: Summary of main existing literature in learning GM.
>
> | Method | Introduction|
> | -------- | ------------------------------------------------------------------------------------------------------------------------------------------------------------------------ |
> |GMN[1]|The seminal work that employs a convolutional neural network to extract node features and constructs an end-to-end model with spectral matching.|
> |PCA-GM[2]| Leveraging intra-graph and cross-graph structural information using graph convolutional networks.|
> |CIE-H[3]|Enhancing end-to-end training by edge embedding and Hungarian-based attention mechanism.|
> |NGMv2[4]|Developing a matching-aware graph convolution scheme with Sinkhorn iteration.|
> |ASAR[5]|An appearance-aware regularizer is employed to explicitly increase the dissimilarities between similar keypoints and improve model robustness through adversarial attacks.|
> |COMMON[6]|Integrating the momentum distillation strategy to balance the quadratic contrastive loss and reduce the impact of bi-level noisy correspondence.|
> * **Second**, we introduce the concept and methods of certified robustness, which aims to design solvers that can verify the consistency of their prediction for any input within a certified perturbation range. As discussed in Sec.2, various methods have been proposed to achieve certified robustness, but only the Randomized Smoothing (RS) class of methods can scale up to large models and ImageNet-level datasets. ***We present some of the representative RS-based methods in Tab.2, along with their applicable scenarios and features.*** We have added the comparison to the paper, please refer to Appendix.B in the updated version for more details. As analyzed in Sec.1, none of these methods can be directly applied to visual GM. Therefore, we propose the CR-OSRS method, which can transform any basic visual GM model (described in Tab.1) into a model with certified robustness.
>
> Table 2: Summary of main existing literature in RS-type methods for robustness certification.
>
> | Method | Introduction|
> | ------ | -------------------------------------------------------------------------------------------------------------------------------------------------------------------------- |
> |RS[7]|A pioneering work on certified robustness for classification tasks, demonstrating that Gaussian smoothing distributions can provide a provable $\ell_2$ perturbation bound.|
> |DSSN[8]|Providing a novel non-additive smoothing robustness certificate for the $\ell_1$ threat model.|
> |Median Smoothing[9]|Developing a new variant of smoothing specifically for detection based on the medians of the smoothed predictions.|
> |RS for Segmentation[10]|Presenting a scalable certification method for image and point cloud segmentation based on randomized smoothing.|
> |RS for Community Detection[11]|Building a new smoothed community detection method via randomly perturbing the graph structure.|
> * **Third**, this work concentrates on the performance of the visual GM model, which only requires visual feature extraction, and not visual recognition. Visual recognition is beyond the scope of this work, so the visual recognition model will not be evaluated and discussed.

---

> ### Author Response · Authors · 2023-11-19
> **Response to Reviewer eczv (2/3)**
>
> >***W2/Q1: The paper lacks some discussion for enlarging its potential impact to other combinatorial tasks or any limitation and difficulty to extend its adaption to other tasks. Have you explored the possibility of extending your approach to address problems beyond graph matching? Given the ubiquity of combinatorial optimization on graphs, a discussion on a potentially more general framework could be beneficial.***
> >
> **R2:** We appreciate your valuable suggestions. Below we will discuss the challenges of extending certified robustness research to general combinatorial optimization (CO) problems. Then we will introduce the particularities of the visual graph matching (GM) problem that this work focuses on and its advancement for the general issue.
> * First, we emphasize that robustness certification aims to provide a mathematical guarantee that the model output is bounded by a certain region when the input is slightly modified. We share your interest in developing a provably robust method that can be applied to a broader range of CO problems. ***However, we encounter several challenges in this direction. First***, different CO problems may have different input and output formats. Therefore, it is necessary to devise a way to unify the diverse input and output formats. For instance, many CO problems can be represented as graph input forms, but there is no standard and unified format for the outputs, to the best of our knowledge. One possible solution is to use a general structured data format to unify the outputs. After unifying the input and output formats, we can design a general robustness certification method. ***Second***, there is currently a lack of sufficient research on the robustness certification of complex models that take graphs as inputs and structured data as outputs. Most of the existing work can only handle simple classification problems. ***Third***, we need to define the notion of certified robustness for CO problems clearly. The paper [12] proposes a heuristic robustness criterion for CO problems, but it does not guarantee the certified robustness of the model.
> * Second, visual GM, as a special case of CO, has a distinctive input and output format, as described in Sec.1. The input consists of pairs of images and their corresponding keypoints, and the output is a matching matrix that indicates the correspondence between the keypoints. To address this task, we propose a novel method that designs a randomized smoothing distribution with correlation constraints and a subspace partitioning. This not only provides a theoretical guarantee for the robustness of visual GM, but also preserves the model performance and the rationality of the verification. In addition, we define the certified robustness of visual GM as the condition that when the perturbation is within the certified input space, the smoothed model always predicts the output within the output subspace. ***These address the above-mentioned challenges of CO robustness certification in the visual GM problem.***
> ***Our work has implications for the robustness certification of general CO problems***. Particularly, our method provides an effective solution for processing matrix output, which is also a special kind of structured data; our proposal of joint smoothing distribution may also be beneficial in inspiring the processing of constrained input formats in many CO problems; the data augmentation and the regularization term may suggest a similar approach to balancing certified robustness and model performance for general CO problems. ***However, the unification of input and output formats, the certification methods for complex graph tasks, and the reasonable certified robustness definitions are still pending to be solved.***
> * In conclusion, we attempt to solve a special case of the robustness certification problem for CO in this paper. We are working on addressing the challenges for general CO certification and will share our findings in the future.

---

> ### Author Response · Authors · 2023-11-19
> **Response to Reviewer eczv (3/3)**
>
> [1] Zanfir A, Sminchisescu C. Deep learning of graph matching[C]//Proceedings of the IEEE conference on computer vision and pattern recognition. 2018: 2684-2693.
>
> [2] Wang R, Yan J, Yang X. Learning combinatorial embedding networks for deep graph matching[C]//Proceedings of the IEEE/CVF international conference on computer vision. 2019: 3056-3065.
>
> [3] Yu T, Wang R, Yan J, et al. Learning deep graph matching with channel-independent embedding and hungarian attention[C]//International conference on learning representations. 2019.
>
> [4] Wang R, Yan J, Yang X. Neural graph matching network: Learning lawler’s quadratic assignment problem with extension to hypergraph and multiple-graph matching[J]. IEEE Transactions on Pattern Analysis and Machine Intelligence, 2021, 44(9): 5261-5279.
>
> [5] Ren Q, Bao Q, Wang R, et al. Appearance and structure aware robust deep visual graph matching: Attack, defense and beyond[C]//Proceedings of the IEEE/CVF Conference on Computer Vision and Pattern Recognition. 2022: 15263-15272.
>
> [6] Lin Y, Yang M, Yu J, et al. Graph matching with bi-level noisy correspondence[C]//Proceedings of the IEEE/CVF International Conference on Computer Vision. 2023: 23362-23371.
>
> [7] Cohen J, Rosenfeld E, Kolter Z. Certified adversarial robustness via randomized smoothing[C]//international conference on machine learning. PMLR, 2019: 1310-1320.
>
> [8] Levine A J, Feizi S. Improved, deterministic smoothing for L_1 certified robustness[C]//International Conference on Machine Learning. PMLR, 2021: 6254-6264.
>
> [9] Chiang P, Curry M, Abdelkader A, et al. Detection as regression: Certified object detection with median smoothing[J]. Advances in Neural Information Processing Systems, 2020, 33: 1275-1286.
>
> [10] Fischer M, Baader M, Vechev M. Scalable certified segmentation via randomized smoothing[C]//International Conference on Machine Learning. PMLR, 2021: 3340-3351.
>
> [11] Jia J, Wang B, Cao X, et al. Certified robustness of community detection against adversarial structural perturbation via randomized smoothing[C]//Proceedings of The Web Conference 2020. 2020: 2718-2724.
>
> [12] Lu H, Li Z, Wang R, et al. ROCO: A General Framework for Evaluating Robustness of Combinatorial Optimization Solvers on Graphs[C]//The Eleventh International Conference on Learning Representations. 2022.

---

### Official Review · Reviewer_9yGa · 2023-12-04

**Soundness:** 2 fair
**Presentation:** 3 good
**Contribution:** 3 good
**Rating:** 5
**Confidence:** 2

**Summary:**

This paper focuses on certifying the robustness of deep visual graph matching and introduces a novel certification method, CR-OSRS. Initially, the method constructs a joint Gaussian distribution and employs a global optimization algorithm to find the optimal distribution. Additionally, during training, the authors enhance model performance through data augmentation with joint Gaussian noise and an output similarity-based regularization term. Finally, two methods, sampling and marginal radii, are proposed for measuring the certified space for quantitative analysis. Experimental results demonstrate that CR-OSRS offers robustness guarantees for visual GM, outperforming direct application of RS.

**Strengths:**

1. The paper provides detailed insights into the four challenges faced by RS and proposing corresponding solutions for each challenge.
2. The introduction of the novel certification method, CR-OSRS, is substantiated with thorough proofs. Moreover, the paper introduces two quantitative metrics, sampling and marginal radii, to measure certified robustness.
3. The experimental results validating the effectiveness of the proposed data augmentation and similarity-based regularization are compelling.

**Weaknesses:**

1. Figure 2 shows that, without employing data augmentation and similarity-based regularization, the performance of CR-OSRS is comparable to RS-GM.
2. Could acceleration be achieved by incorporating entropy regularization into the optimization process?
3. It would be beneficial if the authors could provide an analysis of the computational complexity of this method.
4. The author wants to express too much content in the article, resulting in insufficient details and incomplete content in the main text.
5. The experimental part needs to be reorganized and further improved.

Details comments
1) It is recommended to swap the positions of Sections 4.3 and 4.4. According to the diagram, 4.3 is the training section, and 4.4 aims to measure certified space. Both 4.1 and 4.2 belong to the robustness and testing sections. Therefore, putting these parts together feels more reasonable.
2) The author should emphasize "The article is a general and robust method that can be applied to various GM methods, and we only use NGMv2 as an example." at the beginning of the article, rather than just showing in the title of Method Figure 1. This can better highlight the characteristics and contribution of the method.
3) The experimental part needs to be reorganized and further improved. The experimental section has a lot of content, but the experimental content listed in the main text does not highlight the superiority of the method well, so it needs to be reorganized. Based on the characteristics of the article, the experimental suggestions in the main text should include the following: 1. Robustness comparison and accuracy analysis with other empirical robustness algorithms for the same type of perturbations, rather than just focusing on the RS method, to clarify the superiority of the method. (You should supplement this part.) 2. Suggest using ablation experiments as the second part to demonstrate the effectiveness of the method. 3. Parameter analysis, elucidating the method's dependence on parameters. 4. Consider its applications on six basic algorithms as an extension part. Afterwards, based on the importance, select the important ones to place in the main text, and show the rest in the appendix.
4) In P16, the proof of claim 2, it should be P(I \in B) not P(I \in A).
5) In Table 2 of appendix, the Summary of main existing literature in learning GM can list the related types of perturbations.
6) In Formula 8, please clarify the meaning of lower p (lower bound of unilateral confidence), and the reason and meaning of setting as 1/2.

**Questions:**

1. Can this paper conduct a comparative analysis with [1]?
2. The author proposes a universal robustness framework, and in the experimental part, except RS, other empirical robustness algorithms for the same type of perturbations should be compared to evaluate the advantages of this authentication robustness method.
3. Some details need to be rechecked and explained, as shown in the following.


[1] "InstaBoost++: Visual Coherence Principles for Unified 2D/3D Instance Level Data Augmentation," IJCV

---

### Official Review · Reviewer_mXhz · 2023-12-09

**Soundness:** 3 good
**Presentation:** 3 good
**Contribution:** 2 fair
**Rating:** 5
**Confidence:** 3

**Summary:**

The paper presents a novel method named Certified Robustness based on Optimal Smoothing Range Search (CR-OSRS) to enhance robustness in deep visual graph matching, which is vulnerable to adversarial attacks. This method employs a joint Gaussian distribution for random smoothing, improving structural keypoint matching and balancing robustness with performance. CR-OSRS uses global optimization and similarity-based output space partitioning to manage computational complexity and enhance performance. The effectiveness of the method is demonstrated through experimental results.

**Strengths:**

-The paper proposes CR-OSRS, a method inspired by RS, to build a smoothed model with a joint Gaussian distribution specifically for visual GM application to capture the structural information between keypoints.

-The paper introduces a global optimization algorithm designed to find the optimal parameters for the joint Gaussian distribution, aimed at achieving a larger certified space.

-Applying data augmentation and a similarity-based regularization term during training helps improve the performance of the smoothed model.

**Weaknesses:**

-The paper does not provide a direct comparison between the base model and the smoothed model to support the claim of mitigating performance degradation due to smoothing.

-The improvement in CA is achieved not solely by CR-OSRS but largely through retraining the base model with data augmentation and a regularization term. These additional techniques contribute to the model reaching a similar level of CA at the same radius for RS-GM, too.

-It's unclear how CR-OSRS performs against a wide range of adversarial attacks, particularly those that may not follow the assumptions made in the method's design. E.g, inserting outliers. These outliers could be points that are randomly inserted, or strategically placed by an adversary which can significantly alter the structural information between keypoints.

-The experiments are conducted on a limited number of testing samples for the Pascal VOC and Spair71k datasets. And limited testing samples can affect the generalizability of the results and may not fully demonstrate the method's effectiveness across diverse conditions.

**Questions:**

n/a (emergency review)

---

### Official Review · Reviewer_Gpaf · 2023-12-09

**Soundness:** 2 fair
**Presentation:** 3 good
**Contribution:** 3 good
**Rating:** 6
**Confidence:** 3

**Summary:**

This paper introduces a novel certification method addressing the Visual Graph Matching problem. The authors highlight various challenges associated with applying original randomized smoothing techniques to visual GM. To address these challenges, they design a smoothed model utilizing a joint Gaussian distribution to capture keypoint correlations. Additionally, they implement a similarity threshold to reduce the permutation count. Moreover, they introduce two methods, sampling and marginal radii, to gauge the certified space for paired robustness. The paper offers both theoretical analysis and empirical evaluations of the proposed method.

**Strengths:**

1. The authors tackle several challenges to achieve certified robustness for visual graph matching.

2. The derivation for the certified robustness appears to be sound.

3. The experimental results demonstrate a clear advantage in terms of robustness.

**Weaknesses:**

1. The authors focus on the certified robustness against attacks on keypoint positions in the main content. The investigation of the attacks on node/edge features is limited in this paper. It would be beneficial for the authors to clarify this point.

2. One main contribution of this work is to use the joint Gaussian distribution to build a smoothed model. However, the authors do not clearly elaborate on why they designed $\Sigma$ in this manner. The authors should consider conducting a study on the impact of different construction choices of the joint Gaussian distribution.

3. The authors should compare with more smoothing methods, such as [1] and [2], to demonstrate the advantage of their algorithm.

4. The authors do not demonstrate the matching accuracy of the GM solvers without smoothing. Performance degradation caused by smoothing remains unknown. Therefore, it's hard to tell the significance of the performance improvement in this work, especially considering that the improvement shown in the current numerical results appears to be limited.

5. The presentation of experimental results is unclear, especially when compared with the RS-GM method. The authors could differentiate between different methods by using different line types or thicknesses.

6. In Fig.3(b), the improvement of robustness from s=0.9 to s=0.6 is limited. The authors could test more points between s=1.0 and s=0.9 to demonstrate the results more clearly.

[1] Motasem Alfarra, Adel Bibi, Philip Torr, and Bernard Ghanem. Data dependent randomized smoothing. In The 38th Conference on Uncertainty in Artificial Intelligence, 2022.

[2] Francisco Eiras, Motasem Alfarra, M Pawan Kumar, Philip HS Torr, Puneet K Dokania, Bernard Ghanem, and Adel Bibi. Ancer: Anisotropic certification via sample-wise volume maximization. arXiv preprint arXiv:2107.04570, 2021.

**Questions:**

1. The current design appears strange to me as it seems that for two different nodes $u$ and $v$, the correlation $\Sigma(u,v)$ can be either 0 or $\sigma \cdot b$ depending on their order among all nodes. Is there any reason to design in this way?

2. Since it is no longer a classification problem, why is low bound of the probability in Eq.(7) to be 1/2?

---

### Official Review · Reviewer_gXcF · 2023-12-14

**Soundness:** 2 fair
**Presentation:** 2 fair
**Contribution:** 2 fair
**Rating:** 5
**Confidence:** 3

**Summary:**

This paper proposes an approach to attain certifiable robustness for deep graph
matching (GM) against adversarial inputs. Existing approaches based on
randomized smoothing do not work for GM because of the large output
permutation space. The authors proposed an alternative method called
Certified Robustness based on Optimal Smoothing Range Search (CR-OSRS).
The main idea is to develop a new notion of robustness for permutation
outputs (based on similarity (5)) and to perturb the input keypoint
positions by a joint Gaussian distribution (instead of an isotropic
Gaussian distribution). Methods to choose the joint Gaussian
distribution and to measure the certified space are also developed.

**Strengths:**

1. Robustness of Deep GM is difficult due to the large output permuation
space. This work seems to be the first that aims to attain certifiable
robustness for Deep GM.

**Weaknesses:**

1. I wish the presentation of proposed method is more clear. Right now
both the new robustness notion and the proposed method lack clarity.

**Questions:**

1. The main result appears to be Theorem 4.1, which states that, for
perturbation $(\delta_1, \delta_2)$ of keypoint positions within (9),
the proposed method guarantees that $g_0(...+\delta1, +\delta2)=1$. However, what is the
significance of $g_0(...)=1$? At best, it states that, with another
random perturbation $(\epsilon_1, \epsilon_2)$, the probability that the
output of $f$ is too far from the corresponding core output is higher
than 1/2.  However, it is unclear to me how this claim can connect to
the desirable robustness property, i.e., the output of the deep GM is
robust against the perturbation $(\delta_1, \delta_2)$. Without a clear
explanation of this connection, it is difficult for the reader to
understand the meaning and significance of Theorem 4.1.

2. The construction of $B_1$ and $B_2$ in Section 4.2 also needs better
explanation. I understand that there are non-zero off-diagonal elements
$\sigma \times b$ right adjacent to the main diagonal.  While this does
create correlation between the perturbation of different keypoints, I am lost
why this is a good choice and what purpose it aims to serve. In
particular, these off-diagonal elements correspond to pairs of indices
$(i,i+1)$. However, with arbitrary indexing of the keypoints, $i$ and
$i+1$ may correspond to two keypoints that are far away. Thus, I don't
understand why adding correlation between such a pair of keypoints is a
good idea.

Related to this point, equation (10), which is the objective for
optimizing $B_1$ and $B_2$, also needs more explanation.

3. In Algorithm 2 on page 18 (for optimizing (10)), is Line 11 a
gradient step? How is the gradient calculated?

---

### Meta-Review · Area_Chair_Kt76 · 2023-12-05

**Metareview:**

The paper has received a number of feedbacks from reviewers including several domain experts as emergent reviewers, highlighting several strengths, including rigorous theoretical analysis and proofs for the certified robustness guarantee. The work addresses the challenge of achieving certifiable robustness for Deep GM, representing an interesting effort in this direction. Additionally, the paper introduces two methods to quantify the certified space in visual graph matching. It achieves commendable results on two GM datasets and four GM solvers, showcasing the effectiveness and superiority of the proposed method over the Random Smoothing (RS) method.
AC read this paper himself and SAC also concurs with the below notable weaknesses and areas for improvement:

1.	Reviewers emphasize the need for more explanations in the paper's presentation, covering multiple aspects ranging from graph matching (combinatorial optimization), robustness certification, robustness notion, and details about the optimization algorithm's implementation. Without clear explanations, readers find it hard to follow, as at least three reviewers pointed out.

2.	Reviewers recommend additional study on the impact of different construction choices of the joint Gaussian distribution, the performance degradation caused by smoothing, an investigation of attacks on node/edge features, and more discussion for enlarging its potential impact on other combinatorial tasks.

3.	There is a call for a more thorough experimental comparison and analysis. Specifically, the suggestion is to compare with additional smoothing methods to demonstrate the algorithm's advantages beyond focusing solely on the RS method. Additionally, concerns are raised about the limited number of testing samples for the Pascal VOC and Spair71k datasets, affecting the generalizability of results and potentially limiting the demonstration of the method's effectiveness across diverse conditions.

In summary, while the acknowledged strengths include theoretical analysis, proofs, and some promising results, there are many concerns about paper presentation, investigation, and experimental design and comparison.
In the rebuttal period, AC had asked for additional domain experts to reevaluate this paper but they arrived late. However, late reviews had little influence on our decisions (the meta-review was written before they came) and the new reviews also confirmed our final decision.
Thus, through thorough discussion between AC, SAC and PC, we converge to the point that this paper is not ready for publication and hope these comments could help the authors to improve quality in the next cycle.

**Justification For Why Not Higher Score:**

Reviewers emphasize the need for more explanations in the paper's presentation, covering multiple aspects ranging from graph matching (combinatorial optimization), robustness certification, robustness notion, and details about the optimization algorithm's implementation. Without clear explanations, readers find it hard to follow, as at least three reviewers pointed out.

Reviewers recommend additional study on the impact of different construction choices of the joint Gaussian distribution, the performance degradation caused by smoothing, an investigation of attacks on node/edge features, and more discussion for enlarging its potential impact on other combinatorial tasks.

There is a call for a more thorough experimental comparison and analysis. Specifically, the suggestion is to compare with additional smoothing methods to demonstrate the algorithm's advantages beyond focusing solely on the RS method. Additionally, concerns are raised about the limited number of testing samples for the Pascal VOC and Spair71k datasets, affecting the generalizability of results and potentially limiting the demonstration of the method's effectiveness across diverse conditions.

**Justification For Why Not Lower Score:**

NA.

---

### Decision · Program_Chairs · 2024-01-16

Reject